# Growth, physical, and cognitive function in children who are born HIV-free: School-age follow-up of a cluster-randomised trial in rural Zimbabwe

**Joe D Piper**[1,2,3]*, **Clever Mazhanga**[1], **Marian Mwapaura**[1], **Gloria Mapako**[1], **Idah Mapurisa**[1], **Tsitsi Mashedze**[1], **Eunice Munyama**[1], **Maria Kuona**[1], **Thombizodwa Mashiri**[1], **Kundai Sibanda**[1], **Dzidzai Matemavi**[1], **Monica Tichagwa**[1], **Soneni Nyoni**[1], **Asinje Saidi**[1], **Manasa Mangwende**[1], **Dzivaidzo Chidhanguro**[1], **Eddington Mpofu**[1], **Joice Tome**[1], **Gabriel Mbewe**[1], **Batsirai Mutasa**[1], **Bernard Chasekwa**[1], **Handrea Njovo**[4], **Chandiwana Nyachowe**[4], **Mary Muchekeza**[4], **Kuda Mutasa**[1], **Virginia Sauramba**[1], **Ceri Evans**[1,5], **Melissa J Gladstone**[5], **Jonathan C Wells**[6], **Elizabeth Allen**[3], **Melanie Smuk**[2], **Jean H Humphrey**[7], **Lisa F Langhaug**[1], **Naume V Tavengwa**[1], **Robert Ntozini**[1], **Andrew J Prendergast**[1,2]

**1** Zvitambo Institute for Maternal and Child Health Research, Harare, Zimbabwe, **2** Blizard Institute, Queen Mary University of London, London, United Kingdom, **3** London School of Hygiene and Tropical Medicine, London, United Kingdom, **4** Ministry of Health and Child Care, Harare, Zimbabwe, **5** Institute of Translational Medicine, University of Liverpool, Liverpool, United Kingdom, **6** Population Policy and Practice Research and Teaching Department, UCL Great Ormond Street Institute of Child Health, London, United Kingdom, **7** Department of International Health, Johns Hopkins Bloomberg School of Public Health, Baltimore, Maryland, United States of America

* j.piper@qmul.ac.uk

**Data Availability Statement:** Data will be freely available as individual participant data on ClinEpiDB with an accompanying data dictionary at http://

## Abstract

### Background

Globally, over 16 million children were exposed to HIV during pregnancy but remain HIV-free at birth and throughout childhood by 2022. Children born HIV-free (CBHF) have higher morbidity and mortality and poorer neurodevelopment in early life compared to children who are HIV-unexposed (CHU), but long-term outcomes remain uncertain. We characterised school-age growth, cognitive and physical function in CBHF and CHU previously enrolled in the Sanitation Hygiene Infant Nutrition Efficacy (SHINE) trial in rural Zimbabwe.

### Methods and findings

The SHINE trial enrolled pregnant women between 2012 and 2015 across 2 rural Zimbabwean districts. Co-primary outcomes were height-for-age Z-score and haemoglobin at age 18 months (clinicaltrials.gov NCT01824940). Children were re-enrolled if they were aged 7 years, resident in Shurugwi district, and had known pregnancy HIV-exposure status. From 5,280 pregnant women originally enrolled, 376 CBHF and 2016 CHU reached the trial endpoint at 18 months in Shurugwi; of these, 264 CBHF and 990 CHU were evaluated at age 7 years using the School-Age Health, Activity, Resilience, Anthropometry and Neurocognitive (SAHARAN) toolbox. Cognitive function was evaluated using the Kaufman Assessment

ClinEpiDB.org from 2025. Researchers must agree to the policies and comply with the mechanism of ClinepiDB to access data housed on this platform. Prior to that time, data are available upon reasonable request from the Zvitambo Institute for Maternal and Child Health Research, by contacting Lenin Madhuyu (l.madhuyu@zvitambo.com).

**Funding:** Wellcome Trust [220671/Z/20/Z (JDP), 108065/Z/15/Z (AJP)] https://wellcome.org/; NIH [R61HD103101 (AJP) https://www.nih.gov/]; Thrasher [15250 (AJP)] https://www.thrasherresearch.org/; and IMMANA [3.02 (AJP) https://www.anh-academy.org/immana]. The funders played no role in the study design, data collection, analysis, publication or preparation of this manuscript.

**Competing interests:** The authors have declared that no competing interests exist.

**Abbreviations:** ART, antiretroviral therapy; BIA, Bioimpedance analysis; CBHF, children born HIV-free; CHU, children who are HIV-unexposed; CHW, community health worker; CI, confidence interval; CMV, Cytomegalovirus; DAG, directed acyclic graph; EPDS, Edinburgh Postnatal Depression Score; GEE, generalised estimating equation; HFIAS, household food insecurity assessment scale; IQ, intelligence quotient; IQR, interquartile range; IYCF, infant and young child feeding; MPI, Mental Processing Index; PCR, polymerase chain reaction; PMTCT, prevention of mother-to-child transmission; SD, standard deviation; SHINE, Sanitation Hygiene Infant Nutrition Efficacy; SOC, standard-of-care; SQ-LNS, small quantity lipid-based nutrient supplement; WASH, water, sanitation and hygiene.

Battery for Children (KABC-II), with additional tools measuring executive function, literacy, numeracy, fine motor skills, and socioemotional function. Physical function was assessed using standing broad jump and handgrip for strength, and the shuttle-run test for cardiovascular fitness. Growth was assessed by anthropometry. Body composition was assessed by bioimpedance analysis and skinfold thicknesses. A caregiver questionnaire measured demographics, socioeconomic status, nurturing, child discipline, food, and water insecurity. We prespecified the primary comparisons and used generalised estimating equations with an exchangeable working correlation structure to account for clustering. Adjusted models used covariates from the trial (study arm, study nurse, exact child age, sex, calendar month measured, and ambient temperature). They also included covariates derived from directed acyclic graphs, with separate models adjusted for contemporary variables (socioeconomic status, household food insecurity, religion, social support, gender norms, caregiver depression, age, caregiver education, adversity score, and number of children's books) and early-life variables (length-for-age-Z-score) at 18 months, birthweight, maternal baseline depression, household diet, maternal schooling and haemoglobin, socioeconomic status, facility birth, and gender norms. We applied a Bonferroni correction for the 27 comparisons (0.05/27) with threshold of $p < 0.00185$ as significant. We found strong evidence that cognitive function was lower in CBHF compared to CHU across multiple domains. The KABC-II mental processing index was 45.2 (standard deviation (SD) 10.5) in CBHF and 48.3 (11.3) in CHU (mean difference 3.3 points [95% confidence interval (95% CI) 2.0, 4.5]; $p < 0.001$). The school achievement test score was 39.0 (SD 26.0) in CBHF and 45.7 (27.8) in CHU (mean difference 7.3 points [95% CI 3.6, 10.9]; $p < 0.001$); differences remained significant in adjusted analyses. Executive function was reduced but not significantly in adjusted analyses. We found no consistent evidence of differences in growth or physical function outcomes. The main limitation of our study was the restriction to one of two previous study districts, with possible survivor and selection bias.

## Conclusions

In this study, we found that CBHF had reductions in cognitive function compared to CHU at 7 years of age across multiple domains. Further research is needed to define the biological and psychosocial mechanisms underlying these differences to inform future interventions that help CBHF thrive across the life-course.

## Trial registration

ClinicalTrials.gov The SHINE follow-up study was registered with the Pan-African Clinical Trials Registry (PACTR202201828512110). The original SHINE trial was registered at NCT https://clinicaltrials.gov/study/NCT01824940.

## Author summary

### Why was this study done?

- Over 16 million children globally were born HIV-free (CBHF) to mothers living with HIV by 2022. CBHF have higher mortality, morbidity, and poorer growth and neurodevelopment in early life compared to children who are HIV-unexposed (CHU).

- There is a lack of studies exploring the long-term cognitive, physical and growth outcomes among CBHF compared to children who are HIV-unexposed (CHU).

### What did the researchers do and find?

- The researchers evaluated 264 CBHF and 990 CHU at age 7 years, who had previously been enrolled in the Sanitation Hygiene Infant Nutrition Efficacy (SHINE) cluster-randomised trial in rural Zimbabwe, for a range of cognitive, growth, and physical function outcomes.

- CBHF had reduced cognitive function compared to CHU by up to 0.3 standard deviations (SDs), across a range of cognitive tests, following a Bonferroni correction for multiple comparisons, and adjustment for multiple covariates.

### What do these findings mean?

- CBHF show a reduction in neurodevelopment across multiple cognitive domains, which persists to school age.

- This reduction in cognition in CBHF remains significant even after adjusting for early-life or contemporary social and environmental covariates, suggesting further research is needed to understand the biological and psycho-social drivers of this difference.

- The main study limitations were that CBHF had higher mortality, and may be more likely to relocate, so that measurements may have been influenced by selection bias. Also it was not possible to investigate the impact of different antiretroviral regimens in pregnancy on outcomes among CBHF.

## Introduction

Increasing coverage of prevention of mother-to-child transmission (PMTCT) interventions has advanced progress towards paediatric HIV elimination in sub-Saharan Africa. The vast majority of children born to women with HIV are therefore now HIV-free, meaning they were exposed to HIV in pregnancy but remain uninfected themselves. The current global population of children born HIV-free (CBHF) is increasing, with estimates of 14.8 million in 2018 [1,2] and 15.9 million in 2021 [3]. A disparity in clinical outcomes between CBHF and children who are HIV-unexposed (CHU) emerged in the pre-antiretroviral therapy (ART) era, with

3-fold higher mortality, higher frequency, and severity of common infections, and more growth failure in CBHF [4,5]. Clinical outcomes in the PMTCT era have been uncertain due to a paucity of studies, but emerging data from sub-Saharan Africa confirm that disparities persist in early life despite high coverage of maternal ART. CBHF are more likely to be born premature and small-for-gestational age [6], have a higher frequency of stunting [7], and poorer neurodevelopment [8] than CHU by 2 years of age.

Multiple factors are likely to contribute to these clinical disparities [9] including both universal and HIV-specific risk factors, and the shared mechanisms through which they operate. Early-life biological exposures such as maternal HIV, co-infections, dysbiosis, inflammation, malnutrition, and stress may have persistent effects long after the antenatal exposure has ended. Maternal inflammation is associated with infant mortality among pregnant women living with HIV [10], and intrauterine HIV exposure appears to shape the infant immune system, with a distinct inflammatory milieu, differences in enteropathy, and earlier Cytomegalovirus (CMV) viraemia driving a highly activated and differentiated CD8 T-cell compartment [10]. However, the relative contribution of universal and HIV-specific risk factors in driving clinical differences between CBHF and CHU remains uncertain.

Few studies have conducted long-term follow-up of CBHF to establish whether early-life clinical disparities persist to school age. One study in Zambia found that differences in growth between CBHF and CHU had widened by 7.5 years of age compared to infancy [11], while a study of neurocognitive outcomes across 5 African countries [12] found no differences between CBHF and CHU groups at median age 7 years, but did not evaluate language, which may be most predictive of future function [12]. Other cohorts have reported poorer mathematic abilities [13] and reduced intelligence quotient (IQ) among CBHF in south-east Asia [14], but studies have focused on a limited range of outcomes.

The Sanitation Hygiene Infant Nutrition Efficacy (SHINE) trial in rural Zimbabwe evaluated the effects of improved infant and young child feeding (IYCF) and/or improved water, sanitation and hygiene (WASH) on child stunting and anaemia at 18 months of age. The trial showed that 50% of CBHF were stunted by age 18 months, and that neurodevelopmental scores were lower at 2 years of age among CBHF compared to CHU [15,16]. Here, we report follow-up of a subgroup of CBHF and CHU at age 7 years to evaluate whether disparities in growth, physical, and cognitive function persist to school age [17]. School-age function is highly predictive of later adult function, and therefore represents an important period of mid-childhood. Understanding whether CBHF thrive less well in the long-term is critical to inform the timing of interventions to improve long-term human capital in this expanding global population.

## Methods

### SHINE trial

The design and results of the SHINE trial have been previously reported [18]. In brief, between 2012 and 2015, SHINE recruited 5,280 women during pregnancy from 2 rural Zimbabwean districts with 15% antenatal HIV prevalence (ClinicalTrials.gov NCT01824940). Women who resided in the study area were cluster-randomised to one of 4 intervention arms: standard-of-care (SOC, including promotion of PMTCT and optimal breastfeeding); IYCF (20 g daily small quantity lipid-based nutrient supplement (SQ-LNS) for their infant from 6 to 18 months of age, with complementary feeding counselling); WASH (ventilated improved pit latrine and 2 handwashing stations, monthly liquid soap and chlorine, a play-space to separate children from livestock and to reduce geophagia, plus hygiene counselling); or IYCF plus WASH (all interventions). IYCF improved linear growth and haemoglobin at 18 months in both CBHF and CHU [15,19], while WASH had no effect. The combined IYCF+WASH intervention

improved child neurodevelopment at 2 years of age among CBHF, but neither intervention improved neurodevelopment in CHU [20,21].

### School-age follow-up

To evaluate the long-term effects of HIV exposure on child health outcomes, we designed a substudy to assess child growth, body composition, physical, and cognitive function at 7 years of age. No further trial interventions had been provided after age 18 months. The prespecified research question and analytic approach are outlined in the protocol (objective 3b) and statistical analysis plan (section 5.11) for the follow-up study, which are registered at https://osf.io/8e2zh (accessed 4th May 2024) [22]. Briefly, children were eligible if they were aged 7 to 8 years and still resident in Shurugwi district. Children were ineligible if they were no longer resident in Shurugwi, had an unknown maternal pregnancy HIV status, or were outside the age window. Among all children born to HIV–negative mothers (CHU) and evaluated at the trial endline at age 18 months, 250 per intervention arm meeting the eligibility criteria were randomly selected by computer; those who were unable to be visited or whose family declined participation were replaced by another eligible child randomly selected from the same trial arm. For CBHF, all children in Shurugwi district who were born to HIV–positive mothers and who tested HIV–negative at 18 months were offered enrolment. Families were approached through community health workers (CHWs) to determine if the child was available, and the household was interested in the follow-up study. Written informed consent from the primary caregiver and written assent from the child were obtained by research nurses, following a demonstration of the tools used to conduct the measurements.

### HIV status

Mothers in the SHINE cohort were tested during pregnancy using an antibody rapid test algorithm (Alere Determine HIV-1/2 test, followed by INSTI HIV-1/2 test if positive). Mothers were offered HIV testing at baseline and again at 32 weeks for those who tested negative at baseline; further testing was offered at 18 months postpartum. CHU were defined as children born to mothers testing negative for HIV during pregnancy. Children who were HIV-exposed were defined as those born to mothers testing positive for HIV during pregnancy [21]. CBHF were defined as children who were HIV-exposed and confirmed HIV–negative through 18 months of age (trial endpoint) [15]. The early-life child HIV status was determined by dried blood-spot DNA polymerase chain reaction (PCR), plasma RNA PCR, or rapid test algorithm, depending on child age and sample type.

For the school-age visit, HIV testing was offered to all caregiver-child pairs to ensure that an updated HIV exposure status (in case of maternal incident infection since the end of the trial) and infection status (in case of prolonged breastfeeding and postnatal transmission since the 18-month trial endpoint) was available. If the mother's HIV test was negative, the child was not tested. If the mother tested positive for HIV, declined testing or was unavailable, HIV testing was offered to the child with age-appropriate assent using role plays. The Determine HIV-1/2 rapid test (Abbott) was used for initial testing; positive results were repeated using the HIV 1/2 Stat-Pak rapid test (Chembio). Any children who tested positive for HIV were referred to local clinics for ART and were excluded from this analysis. All assessments were done by study nurses who were blinded to the caregiver and child HIV status.

### School-age measurements using the SAHARAN toolbox

We developed a battery of tests to measure growth, physical, and cognitive outcomes, termed the School-Age Health, Activity, Resilience, Anthropometry and Neurocognitive (SAHARAN)

toolbox, which has been previously published [23]. Measurements were performed during a single home visit by extensively trained and supervised primary care nurses, using 1 or 2 tents pitched in the household or nearby. Briefly, the SAHARAN toolbox comprises a caregiver questionnaire, child questionnaire, and direct tests undertaken with the child to assess cognitive function, growth, and physical function. All tests had standardised explanations, demonstrations, and translations in local languages (Shona and Ndebele). Data collectors underwent standardisations in anthropometry and cognition tests every 9 months and had regular supported supervision.

Cognitive function was assessed using a range of tests administered in the same order each time. All tests had been validated in a pilot study of 80 children not in the SHINE trial [23]. Firstly, a custom-built School Achievement Test (SAT) was administered, which measured literacy and numeracy, followed by the Kaufman Assessment Battery for Children 2nd edition (KABC-II; Pearson UK) which was locally adapted as previously published [24,25]. Next, a coordination test was undertaken [26] which measured fine motor skills by finger tapping. Finally for cognition, 3 subtests from the android tablet-based Plus Executive Function test (Plus-EF) [27] were administered to measure executive function. The child's socioemotional function and behaviour were measured by the caregiver-reported Strengths and Difficulties Questionnaire (SDQ) https://www.sdqinfo.org/ (accessed 4th May 2024). An overall score of the child's functional abilities was provided by the Washington Group/UNICEF screening tool [28], which helped identify underlying physical disabilities in sight, hearing and mobility, or behavioural and learning difficulties.

After cognition, growth measurements were performed in the same order as previously piloted [23] including height, knee-heel length, weight, head circumference, mid-upper arm circumference (MUAC), and waist, hip and calf circumferences. Body composition was assessed using Holtain calipers (Crosswell, UK) to measure peripheral subcutaneous fat (triceps and calf skinfold thicknesses) and central subcutaneous fat (subscapular and supra-iliac skinfold thicknesses). Bioimpedance analysis (BIA) using a BodyStat 1500 MDD machine (BodyStat, Isle of Man, UK) assessed lean mass, measured as the impedance index (height$^2$/impedance) and lean mass index (1/impedance) [29]; and tissue health, measured as the phase angle.

Physical function was then measured as previously described [23]. Leg strength was measured by the distance jumped in the broad jump from a standing position, and handgrip strength in each hand using a dynamometer (Takei, Japan). Finally, cardiovascular fitness was measured by the shuttle-run test, where the child repeatedly ran between 2 markers placed 20 metres apart, arriving at each end before a timed beep from an android app (Beep Test, Ruval Enterprises, Canada) connected to a Bluetooth speaker. The child ran until either they missed the beep 3 times in a row or stopped due to tiredness. Maximal Oxygen consumption (VO$_2$max) was calculated, which represents the maximum rate at which the body uses oxygen during exercise, [30]. Resting and post-exercise blood pressure was measured using a manual aneroid sphygmomanometer (Medisave, UK). Haemoglobin was measured (Hemocue) on a finger-prick blood sample.

A caregiver questionnaire was used to record contemporary household demographics, socioeconomic status using a locally validated wealth index [31], schooling exposure, adversities, nurturing (including the Child-Parent Relationship scale [32] and the Multi-Indicator Cluster Survey (MICS) Child Discipline questionnaire [33,34]), caregiver depression [35], gender norms [36], caregiver social support [36], and food [37,38], and water [39] insecurity.

Most data were collected electronically using Open Data Kit (ODK, getodk.org; accessed 29 April 2024) on android tablets (Samsung Galaxy Tab A). The KABC-II and SAT used paper forms subsequently transcribed onto ODK. The Plus-EF tool recorded data directly into the Plus-EF application.

## Statistical analysis

A prespecified statistical analysis plan is available at https://osf.io/8e2zh (accessed 5 May 2024). Stata v13 and v17 were used for analyses. Baseline characteristics between CBHF and CHU groups were compared using multinomial and ordinal regression models and Somers' D for medians, while handling within-cluster correlation with robust variance estimation. Generalised estimating equations (GEEs) were used to compare each functional outcome between CBHF and CHU groups. The first analysis estimated unadjusted differences between CBHF and CHU. Then, 3 separate adjusted analyses were performed: Model 1 adjusted for trial factors: randomised intervention arm, study nurse performing the assessment, exact age of child, sex of child, and calendar season the child was measured. Model 2 included these trial factors plus contemporary socioeconomic and demographic confounders to explore the impact of the child's current environment on growth and function. These confounders were measured during the caregiver questionnaire at the 7-year visit and identified from a directed acyclic graph (DAG) with online software (Dagitty.net; accessed 5 May 2024, see Supporting information, S1 and S2 Figs); DAGitty eliminates colliders and mediators in selecting the covariates. For the contemporary model, confounders included socioeconomic status, caregiver depression (Edinburgh Postnatal Depression Score, EPDS), household food insecurity assessment scale (HFIAS), household religion, caregiver social support [40], caregiver gender norms [40], caregiver age, caregiver education, adversity score, and the number of children's books at home. Model 3 adjusted for the trial factors and early-life covariates identified from a separate DAG (S2 Fig), constructed to explore the impact of the early-life environment on school-age child growth and function. These included socioeconomic and demographic confounders, measured during the SHINE baseline visit shortly after the mother's enrolment during pregnancy. These included baseline EPDS, household dietary diversity score, maternal haemoglobin, household socioeconomic score, maternal education, and maternal gender norms. Early-life child-based covariates also included in this model were health facility births, birthweight, and length-for-age score at 18 months. Since there were 27 outcomes, a Bonferroni correction was applied (0.05/27) leading to a threshold of $p < 0.00185$ being considered significant (referred to as $p < 0.002$ subsequently). A prespecified subgroup analysis was also performed if there was evidence for an interaction of CBHF with child sex ($p < 0.10$).

## Ethics

The Medical Research Council of Zimbabwe approved the study protocol (MRCZ/A/1675). The SHINE follow-up study was registered with the Pan-African Clinical Trials Registry (PACTR202201828512110). Written informed consent from the primary caregiver and written assent from the child were obtained, following a demonstration of the tools used to conduct the measurements.

## Results

Between 22 November 2012 and 27 March 2015, 5,280 pregnant women were enrolled from 211 clusters at median 12 (interquartile range, IQR 9, 16) gestational weeks in Chirumanzu and Shurugwi districts (Fig 1). In Shurugwi, there were 420 births to women with HIV and 2,174 births to women without HIV; 376 HIV-exposed and 2016 HIV-unexposed children completed the 18-month primary endpoint visit. Between 18 months and 7 years, 2 (0.5%) CBHF and 5 (0.2%) CHU died, while 6 (1.6%) CBHF and 42 (2.1%) CHU were lost to follow-up. Two caregivers of CBHF and 9 caregivers of CHU declined follow-up at 7 years. There were 87 (23.2%) relocations among CBHF and 387 (19.2%) relocations among CHU; these children were therefore ineligible for inclusion. Overall, 273 HIV-exposed children were

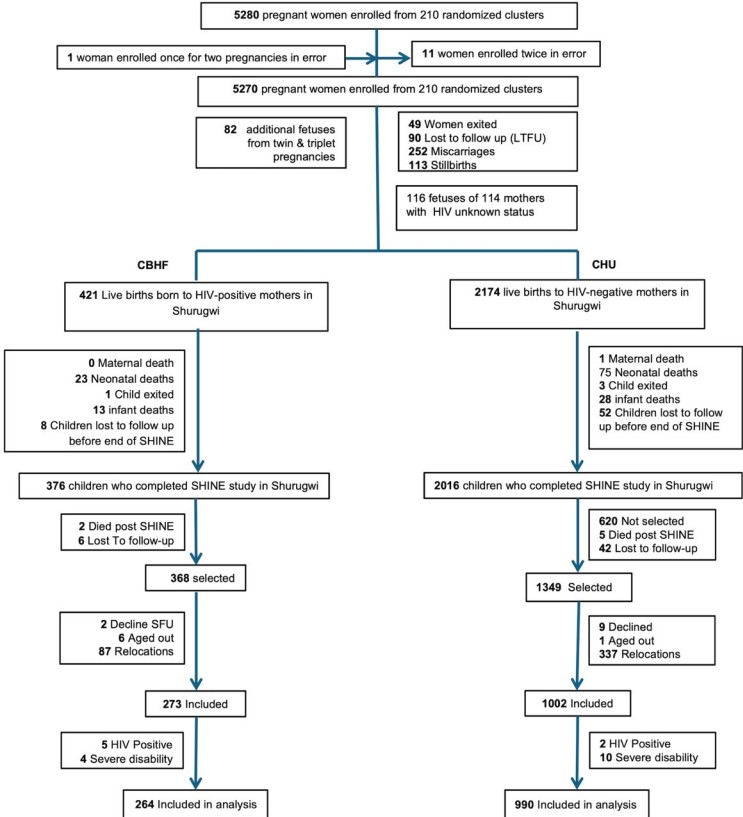

**Fig 1. CONSORT diagram showing CBHF and CHU selected into the SHINE trial follow-up (abbreviated to SFU).** Note that 6 children born to women with HIV aged out because initially children with known HIV–positive status were not included for measurement; 1 child born to a mother without HIV also aged out due to heavy rains making their area inaccessible before they turned 8 years old. Participants were recorded as LTFU at 3 stages: shortly after enrolment as pregnant mothers, during the first 18 months before the trial primary endpoint was measured, or between 18 months and 7 years. Children with severe disability or who were living with HIV were not included in the current analysis. CBHF, children born HIV-free; CHU, children who are HIV-unexposed; SHINE, Sanitation Hygiene Infant Nutrition Efficacy.

measured, of whom 5 were HIV–positive and 4 had severe disability, leaving 264 CBHF in the current analysis. Of 1,002 HIV-unexposed children measured, 12 were subsequently excluded: 2 were HIV–positive (due to mothers seroconverting during breastfeeding) and 10 had severe disability; 990 CHU were included in the current analysis.

## Participant characteristics

Characteristics of the child, caregiver, and household at the time of the follow-up visit are shown in Table 1. CBHF compared to CHU had a higher proportion of caregivers who were mothers (83.1% versus 75.9%) compared to other types of caregiver such as grandmothers or other family members. CBHF also had a greater proportion who had been randomised to WASH or combined arms. CBHF households had higher levels of food insecurity as measured by the HFIAS score (12.9 versus 12.0) and a higher adversity score (1.95 versus 1.77) compared to CHU households. Caregivers of CBHF versus CHU had slightly fewer years of schooling (9.5 versus 10.0), although both were relatively high. CBHF caregivers also had higher

**Table 1. Contemporary characteristics of CBHF and CHU at the 7-year follow-up visit.**

| Variable at 7 –year visit | CBHF n = 264 | CHU n = 990 | p-Value | Statistical test |
|---|---|---|---|---|
| **Proportion female; n (%)** | 135/264 (51.1%) | 506/990 (51.1%) | 0.99 | Chi-Sq |
| **Child age, years; mean (SD)** | 7.3 (0.3) | 7.3 (0.2) | 0.07 | GEE (robust) |
| **Mean years of schooling (SD)** | 3.1 (0.7) | 3.3 (0.8) | 0.01 | GEE (robust) |
| **Caregiver characteristics** | | | | |
| **Mother as caregiver; n (%)** | 219/264 (83.3%) | 750/990 (75.8%) | 0.01 | Chi-Sq |
| **Mean years of schooling (SD)** | 9.5 (2.7) | 10.0 (2.6) | 0.01 | GEE (robust) |
| **Mean EPDS (SD)** | 4.0 (4.7) | 3.2 (4.4) | 0.01 | GEE (robust) |
| **Mean social support score (SD)** | 3.8 (0.6) | 3.9 (0.5) | 0.38 | GEE (robust) |
| **Mean gender norms attitudes (SD)** | 4.1 (0.6) | 4.1 (0.6) | 0.62 | GEE (robust) |
| **Mean total discipline score (SD)** | 2.0 (2.1) | 1.9 (2.0) | 0.62 | GEE (robust) |
| **Mean CPRS (SD)** | 3.3 (0.8) | 3.3 (0.7) | 0.21 | GEE (robust) |
| **Household characteristics** | | | | |
| **Socioeconomic status (SES) quintile, n (%)** | | | | |
| Lowest | 62/262 (23.7%) | 186/974 (19.1%) | 0.50 | Chi-Sq |
| Second | 53/262 (20.2%) | 193/974 (19.8%) | | |
| Middle | 50/262 (19.1%) | 197/974 (20.2%) | | |
| Fourth | 51/262 (19.5%) | 197/974 (20.2%) | | |
| highest | 46/262 (17.6%) | 201/974 (20.6%) | | |
| **Mean SES scale (SD)** | 1.5 (0.7) | 1.6 (0.6) | 0.06 | GEE (robust) |
| **Mean HFIAS (SD)** | 12.9 (4.7) | 12.0 (4.2) | 0.005 | GEE (robust) |
| **Mean HDDS (SD)** | 7.7 (2.0) | 7.7 (1.8) | 0.91 | GEE (robust) |
| **Mean Total Household Water Insecurity Experiences scale (HWISE) (SD)** | 12.2 (1.3) | 12.1 (0.9) | 0.41 | GEE (robust) |
| **Female headed household, % [n]** | 18.6% [50] | 17.1% [171] | 0.56 | Chi-Sq |
| **Mean Adversity Score (SD)** | 1.9 (1.5) | 1.8 (1.4) | 0.07 | GEE (robust) |
| **Median number of children's books at home (IQR)** | 0 (0, 1) | 0 (0, 1) | 0.02 | Somers' D |
| **Original trial arm, n (%)** | | | | |
| SOC | 51/264 (19.3%) | 246/990 (24.8%) | <0.001 | Chi-Sq |
| IYCF | 46/264 (17.4%) | 250/990 (25.3%) | | |
| WASH | 93/264 (35.2%) | 247/990 (24.9%) | | |
| IYCF and WASH | 74/264 (28.0%) | 247/990 (24.9%) | | |

Variables all measured at the time of the follow-up visit. Baseline factors measured during participation in the original trial are shown in Table B in S1 Text [23]. The EPDS, Social support, Gender norms attitudes, Total Discipline, CPRS, SES, HFIAS, HDDS, HWISE and Adversity scales are further explained in supplementary materials. %: percentage, n: number of participants, SD: standard deviation, IQR: interquartile range, SOC: Standard of Care, SHINE: Sanitation Hygiene Infant Nutrition Efficacy trial intervention arm, IYCF: Infant and young child feeding (nutrition) arm, WASH: Water, sanitation and hygiene arm, IYCF & WASH: Combined IYCF and WASH arm, Chi-Sq: Chi-Squared test from logistic regression, adjusted for clustering. GEE (robust) Generalised estimating equations with robust variance estimation adjusted for clustering, Somers' D comparison of medians using t-distribution adjusted for clustering. CBHF, children born HIV-free; CHU, children who are HIV-unexposed.

depression scores (4.1 versus 3.2) compared to CHU. CBHF also had lower total schooling exposure (3.1 versus 3.3 years).

Baseline factors measured during participation in the original trial are shown in Tables A and B in S1 Text. Mothers with HIV had lower haemoglobin and MUAC, and higher parity than mothers without HIV, together with higher depression and food insecurity scores, and lower socioeconomic scores. In pregnancy, women with HIV had a mean (standard deviation, SD) CD4 count of 474 (215); 83% were receiving ART, which was predominantly tenofovir-

based regimens (70%). Between birth and 18 months, CBHF had lower anthropometry, with higher rates of stunting and a lower breastfeeding duration, compared to CHU (see Tables A and B in S1 Text).

Early-life reductions in growth for CBHF were also noted in this follow-up study (Table C in S1 Text), as previously reported [7]. At 18 months, CBHF in this follow-up study had lower mean (SD) length-for-age Z-score (LAZ) of −1.83 (1.06) compared to mean (SD) among CHU of −1.50 (1.02). At 18 months, CBHF compared to CHU had a higher proportion of under-weight (16.9% versus 8.8%). CBHF also had a smaller mean (SD) head circumference for age Z-score of −0.46 [1.07] compared to CHU (−0.21 [0.98]) at 18 months.

## Cognitive, physical, and growth outcomes

Cognitive outcomes for CBHF and CHU are shown in Table 2. CBHF had lower total neuro-developmental scores, as measured by the Mental Processing Index from the KABC-II test, which reflects overall cognitive function. There was still strong evidence of difference after adjustment for contemporary or baseline factors. CBHF also had lower scores on the School Achievement Test including in adjusted models. CBHF had reduced executive function as measured by the Plus-EF score in unadjusted models only. There was weak evidence that fine motor function was lower among CBHF in unadjusted models, but not after adjustment for

**Table 2. Cognitive function in CBHF and CHU.**

| Outcome | CBHF | | CHU | | GEE mean difference (95% CI) | | | | | | |
|---|---|---|---|---|---|---|---|---|---|---|---|
| Cognitive variables | N | Mean (SD) | N | Mean (SD) | Unadjusted difference ($p < 0.002$) | | Adjusted difference Model 1 (Trial factors) | | Adjusted difference Model 2 (Trial factors and contemporary covariates) | | Adjusted difference Model 3 (Trial factors and baseline covariates) |
| | | | | | | p | | p | | p | | p |
| **Mental Processing Index** | 264 | **45.2 (10.5)** | 990 | **48.3 (11.3)** | **3.3 [2.0, 4.5]** | **<0.001** | **2.8 (1.6, 4.1)** | **<0.001** | **2.3 (1.1, 3.5)** | **<0.001** | **2.7 (1.5, 4.0)** | **<0.001** |
| **School Achievement Test** | 264 | **39.0 (26.0)** | 990 | **45.7 (27.8)** | **7.3 (3.6, 10.9)** | **<0.001** | **6.8 (3.1, 10.5)** | **<0.001** | 5.1 (1.6, 8.6) | 0.005 | **6.1 (2.4, 9.9)** | **0.001** |
| **Plus executive function (Plus-EF) score** | 251 | **109.4 (24.7)** | 978 | **114.4 (24.2)** | **5.4 (2.3, 8.5)** | **0.001** | **5.3 (2.0, 8.5)** | **0.001** | 4.2 (1.0, 7.4) | 0.010 | 4.8 (1.5, 8.1) | 0.005 |
| **Fine motor speed, seconds** | 262 | 25.0 (6.7) | 986 | 24.1 (6.6) | −1.0 (−1.9, −0.2) | 0.022 | −0.8 (−1.7, 0.1) | 0.074 | −0.5 (−1.4, 0.4) | 0.246 | −0.7 (−1.6, 0.2) | 0.197 |
| **Strengths and Difficulties Questionnaire (SDQ) score** | 263 | 9.1 (5.3) | 989 | 8.6 (5.2) | −0.5 (−1.2, 0.2) | 0.178 | −0.5 (−1.2, 0.2) | 0.188 | −0.1 (−0.9, 0.6) | 0.699 | −0.5 (−1.2, 0.2) | 0.197 |
| **Child socioemotional score** | 256 | 3.7 (0.7) | 973 | 3.7 (0.7) | 0.0 (−0.1, 0.1) | 0.624 | 0.0 (0.0, 0.0) | 0.529 | 0 (−0.1, 0.1) | 0.648 | 0 (−0.1, 0.1) | 0.489 |

Effect sizes shown with 95% Confidence intervals (CI). SD: standard deviation, GEE: Generalised estimating equations with exchangeable correlation structure, used to calculate the difference. Model 1 is adjusted for trial factors (arm, study nurse, exact child age, calendar month recruited, temperature, sex). Model 2 is adjusted for trial factors from Model 1 and contemporary factors (socioeconomic status, caregiver depression score (EPDS), household food insecurity (HFIAS), household religion, caregiver social support, caregiver gender norms, caregiver age, caregiver education, adversity score, children's books at home). Model 3 is adjusted for trial factors from Model 1 and early-life factors (length for age Z-score (LAZ) at 18 mo, birthweight, maternal baseline depression score (EPDS), household diet, maternal haemoglobin, socioeconomic status, facility birth, gender norms, and maternal years of schooling). Overall, 25 Plus-EF measurements were missing due to a programming error on encryption which led to some results being lost. For fine motor assessments, 6 children did not perform the task: 1 child's caregiver refused, and 5 children were unable to fully understand or concentrate for the finger tapping task. Two caregivers did not answer all SDQ questions. Overall, 25 children refused to answer all questions on the child socioemotional scale, hence were unable to provide a full score. A Bonferroni correction was applied for the 27 comparisons, with a $p$-value <0.00185 considered significant; results reaching this threshold are highlighted in bold. CBHF, children born HIV-free; CHU, children who are HIV-unexposed.

**Table 3. Physical function in CBHF and CHU.**

| Outcome | | CBHF | | CHU | GEE mean difference (95% CI) | | | | | | | |
|---|---|---|---|---|---|---|---|---|---|---|---|---|
| Physical function variables | N | Mean (SD) | N | Mean (SD) | Unadjusted difference | | Adjusted difference Model 1 (Trial factors) | | Adjusted difference Model 2 (Trial factors and contemporary covariates) | | Adjusted difference Model 3 (Trial factors and baseline covariates) | |
| | | | | | | p | | p | | p | | p |
| **Grip Strength, Kg** | 262 | 10.5 (1.9) | 990 | 10.7 (1.9) | 0.2 (−0.1, 0.5) | 0.160 | 0.3 (0.0, 0.5) | 0.041 | 0.2 (0.0, 0.5) | 0.062 | 0.3 (0, 0.5) | 0.070 |
| **Broad jump, cm** | 259 | 111.0 (17.3) | 987 | 112.8 (15.1) | 2.1 (−0.2, 4.3) | 0.067 | 2.5 (0.4, 4.6) | 0.022 | 1.8 (−0.4, 4.1) | 0.107 | 1.9 (−0.3, 4) | 0.088 |
| **VO$_2$max, ml kg$^{-1}$ min$^{-1}$** | 255 | 50.2 (2.7) | 986 | 50.9 (2.7) | **0.8 (0.4, 1.2)** | **<0.001** | 0.6 (0.2, 1.0) | 0.006 | 0.5 (0.1, 0.9) | 0.016 | 0.5 (0.1, 0.9) | 0.023 |
| **Systolic BP, mm Hg** | 264 | 96.7 (9.0) | 988 | 97.0 (9.3) | 0.1 (−1.1, 1.2) | 0.279 | 0.5 (−0.5, 1.6) | 0.941 | 0.2 (−0.8, 1.3) | 0.653 | 0.0 (−0.9, 1) | 0.937 |
| **Diastolic BP, mm Hg** | 264 | 62.8 (7.3) | 988 | 62.3 (7.5) | −0.5 (−1.5, 0.4) | 0.889 | 0.0 (−0.9, 1.0) | 0.333 | −0.1 (−1.0, 0.) | 0.746 | 0.6 (−0.5, 1.6) | 0.317 |

Effect sizes shown with 95% confidence intervals (CI). SD: Standard deviation, BP: Blood pressure, VO$_2$max: The maximal oxygen consumption calculated from the level achieved in the shuttle run test. GEE: Generalised estimating equations with exchangeable correlation structure, used to calculate the difference for all variables. Model 1 is adjusted for trial factors (arm, study nurse, exact child age, calendar month recruited, temperature, sex). Model 2 is adjusted for trial factors from Model 1 and contemporary factors (socioeconomic status, caregiver depression score (EPDS), household food insecurity (HFIAS), household religion, caregiver social support, caregiver norms, caregiver age, caregiver education, adversity score, children's books at home). Model 3 is adjusted for trial factors from Model 1 and early-life factors (length for age Z-score (LAZ) at 18 mo, birthweight, maternal baseline depression score (EPDS), household diet, maternal haemoglobin, socioeconomic status, facility birth, gender norms, and maternal years of schooling). Two children (both CBHF) did not perform the grip strength test: 1 was not motivated and 1 had a caregiver who refused the measurements. Eight children did not perform the broad jump test for a variety of reasons: 1 caregiver refused, 1 visit was performed during heavy rains and the ground was too slippery, 2 children had painful legs, 2 children's caregivers refused due to the child being known to have asthma, 1 child refused and 1 measurement was missing without a recorded reason. In addition, 5 children did not do the shuttle run test: 3 children refused, 1 was recorded as having asthma, and 1 measurement was missing without a recorded reason. Two children refused blood pressure measurements. A Bonferroni correction was applied for the 27 comparisons, with a *p*-value <0.00185 considered significant; results reaching this threshold are highlighted in bold. CBHF, children born HIV-free; CHU, children who are HIV-unexposed.

contemporary or baseline factors. There was no evidence of difference between groups in the Strengths and Difficulties Questionnaire or total socioemotional score. Taken together, CBHF showed 0.2–0.3 standard deviation reductions in neurodevelopmental scores across a range of measures of cognition and academic achievement, with some evidence of reduced executive function, particularly in unadjusted models at 7 years.

Physical function scores for both groups are shown in Table 3. While CBHF generally had lower scores for all tests, the only test with strong evidence of difference between groups was the level achieved in the shuttle run test, which reflects VO$_2$max as a measure of cardiovascular fitness. CBHF compared to CHU had a 0.8 ml kg$^{-1}$ min$^{-1}$ lower VO$_2$max, representing 0.3 lower average level (or approximately 60 metres shorter distance) on the shuttle run test, which was significant in unadjusted models only at age 7 years. Growth and body composition for both groups are shown in Table 4. CBHF generally had lower growth and body composition values than CHU, but there was only strong evidence of difference in head circumference, which was 0.3 cm lower in CBHF, across all models at age 7 years. Taken together, CBHF generally had lower physical fitness scores and reduced growth, with strong evidence of difference for cardiovascular fitness and head circumference.

## Subgroup analysis

Out of 27 comparisons, 4 provided evidence of an interaction. Among cognitive outcomes, there was evidence of an interaction between sex of children and the Strengths and Difficulties

**Table 4. Growth and body composition in CBHF and CHU.**

| Outcome | CBHF | | CHU | | GEE mean difference (95% CI) | | | | | | | |
|---|---|---|---|---|---|---|---|---|---|---|---|---|
| Growth and body composition variables | N | Mean (SD) | N | Mean (SD) | Unadjusted difference | | Adjusted difference Model 1 (Trial factors) | | Adjusted difference Model 2 (Trial factors and contemporary covariates) | | Adjusted difference Model 3 (Trial factors and baseline covariates) | p |
| | | | | | | p | | p | | p | | p |
| Height-for-age Z score | 263 | −0.6 (0.9) | 990 | −0.5 (0.9) | 0.1 (0.0, 0.3) | 0.018 | 0.1 (0.0, 0.3) | 0.010 | 0.1 (0.0, 0.2) | 0.035 | 0.1 (0.0, 0.2) | 0.060 |
| Weight-for-age Z score | 262 | −0.7 (0.9) | 988 | −0.6 (0.9) | 0.1 (0.0, 0.2) | 0.228 | 0.1 (0.0, 0.2) | 0.064 | 0.1 (0.0, 0.2) | 0.09 | 0.1 (0.0, 0.2) | 0.154 |
| Body mass index, kg/m$^2$ | 262 | −0.5 (0.9) | 988 | −0.5 (0.8) | 0.0 (−0.2, 0.1) | 0.577 | 0.0 (−0.1, 0.1) | 0.976 | 0.0 (−0.1, 0.1) | 0.946 | 0.0 (−0.1, 0.1) | 0.963 |
| Knee-heel length, cm | 262 | 37.3 (2.0) | 989 | 37.4 (1.9) | 0.1 (−0.1, 0.4) | 0.410 | 0.2 (0.0, 0.5) | 0.068 | 0.2 (0, 0.4) | 0.120 | 0.2 (−0.1, 0.4) | 0.194 |
| Head circumference, cm | 263 | 51.0 (1.5) | 990 | 51.3 (1.4) | 0.3 (0.1, 0.5) | 0.009 | 0.3 (0.1, 0.5) | 0.002 | 0.3 (0.1, 0.4) | 0.005 | 0.3 (0.1, 0.5) | 0.002 |
| Mid-upper arm circumference, cm | 262 | 17.0 (1.4) | 989 | 16.9 (1.3) | −0.1 (−0.3, 0.1) | 0.318 | 0.0 (−0.2, 0.2) | 0.917 | 0.0 (−0.2, 0.1) | 0.596 | 0.0 (−0.2, 0.1) | 0.708 |
| Waist circumference, cm | 263 | 54.1 (3.1) | 989 | 54.1 (3.1) | −0.1 (−0.6, 0.3) | 0.561 | 0.1 (−0.3, 0.5) | 0.753 | 0.1 (−0.3, 0.5) | 0.722 | 0.0 (−0.4, 0.4) | 0.931 |
| Hip circumference, cm | 263 | 61.0 (4.1) | 990 | 60.9 (3.9) | −0.1 (−0.7, 0.5) | 0.790 | 0.1 (−0.5, 0.7) | 0.629 | 0.1 (−0.5, 0.6) | 0.809 | 0.1 (−0.5, 0.6) | 0.837 |
| Calf circumference, cm | 263 | 23.3 (1.6) | 989 | 23.4 (1.7) | 0.1 (−0.1, 0.3) | 0.355 | 0.2 (0.0, 0.4) | 0.074 | 0.2 (−0.1, 0.4) | 0.135 | 0.2 (−0.1, 0.4) | 0.129 |
| Lean mass index, Ohms$^{-1}$ | 262 | 12.1 (1.3) | 982 | 12.1 (1.3) | 0.0 (−0.2, 0.2) | 0.833 | 0.0 (−0.1, 0.2) | 0.678 | 0.0 (−0.2, 0.2) | 0.812 | 0.0 (−0.1, 0.2) | 0.686 |
| Impedance Index, m$^2$/Ohms$^{-1}$ | 262 | 1.7 (0.3) | 982 | 1.8 (0.3) | 0.0 (0.0, 0.1) | 0.225 | 0.0 (0.0, 0.1) | 0.046 | 0.0 (0.0, 0.1) | 0.102 | 0.0 (0.0, 0.1) | 0.111 |
| Phase angle, degrees | 261 | 5.1 (0.5) | 982 | 4.9 (0.6) | −0.1 (−0.2, 0.0) | 0.002 | −0.1 (−0.0.1, 0) | 0.014 | −0.1 (−0.2, 0.0) | 0.019 | −0.1 (−0.1, 0.0) | 0.023 |
| Total skinfold thicknesses, mm | 261 | 26.4 (6.0) | 987 | 27.1 (6.2) | 0.6 (−0.3, 1.5) | 0.182 | 0.7 (−0.2, 1.6) | 0.125 | 0.6 (−0.3, 1.5) | 0.208 | 0.7 (−0.2, 1.5) | 0.137 |
| Peripheral skinfold thickness, mm | 261 | 15.7 (3.6) | 988 | 16.2 (3.7) | 0.5 (0.0, 1.0) | 0.051 | 0.5 (0.0, 1.0) | 0.058 | 0.4 (−0.1, 0.9) | 0.097 | 0.5 (0.0, 1.0) | 0.053 |
| Central skinfold thickness, mm | 263 | 10.9 (3.1) | 989 | 10.9 (3.1) | 0.1 (−0.4, 0.5) | 0.821 | 0.2 (−0.3, 0.6) | 0.539 | 0.1 (−0.4, 0.6) | 0.68 | 0.1 (−0.4, 0.6) | 0.641 |
| Haemoglobin, g dl$^{-1}$ | 261 | 12.6 (1.3) | 990 | 12.7 (1.2) | 0.0 (−0.1, 0.2) | 0.635 | 0.1 (−0.1, 0.2) | 0.547 | 0.1 (−0.1, 0.3) | 0.443 | 0.1 (−0.1, 0.2) | 0.508 |

SD: Standard deviation, GEE: Generalised estimating equations with exchangeable correlation structure, used to calculate the difference for all variables. Model 1 is adjusted for trial factors (arm, study nurse, exact child age, calendar month recruited, temperature, sex). Model 2 is adjusted for trial factors from Model 1 and contemporary factors (socioeconomic status, caregiver depression score (EPDS), household food insecurity (HFIAS), household religion, caregiver social support, caregiver gender norms, caregiver age, caregiver education, adversity score, children's books at home). Model 3 is adjusted for trial factors from Model 1 and early-life factors (length for age Z-score (LAZ) at 18 mo, birthweight, maternal baseline depression score (EPDS), household diet, maternal haemoglobin, socioeconomic status, facility birth, gender norms, and maternal years of schooling). One caregiver refused all anthropometry measurements. Five children refused skinfold measurements, 3 children had a missing weight due to faulty weighing scales, 9 children had missing bioimpedance measurements because of faulty machines or measurements that were excluded for inconsistency, 2 children refused haemoglobin measurements, and 1 child had missing knee-heel length, mid-upper arm circumference (MUAC), waist and calf circumference measurements. A Bonferroni correction was applied for the 27 comparisons, with a *p*-value <0.00185 considered significant; results reaching this threshold are highlighted in bold. CBHF, children born HIV-free; CHU, children who are HIV-unexposed.

questionnaire: among boys, CHU scored better than CBHF (1 mark, [95% CI 0.3, 2.3], $p = 0.032$) but among girls, there was no evidence of difference in SDQ score between CBHF and CHU (Table D in S1 Text). There was evidence of an interaction for the child's socioemotional score ($p = 0.050$), but differences were very small. Among physical function outcomes, there was evidence of an interaction between child sex and $VO_2$max, whereby boys in the CHU group had better cardiovascular fitness than boys born HIV-free (1.2, [95% CI 0.6, 1.8], $p = 0.092$), while there was no evidence of difference between groups among girls. For growth outcomes, there was evidence of an interaction between sex and calf circumference such that girls in the CHU group had weak evidence for a slightly larger calf circumference (0.3 cm, [95% CI 0.0, 0.6], $p = 0.052$) but there was no evidence of a difference for boys. There were no significant interactions with sex of the children for other growth or body composition outcomes.

## Discussion

The population of children born HIV-free continues to expand due to improved coverage of PMTCT interventions among women living with HIV. It is now apparent that, despite ART during pregnancy, CBHF in sub-Saharan Africa have poorer early-life growth [7] and neurodevelopment [41]; however, few studies have evaluated child outcomes beyond 2 years of age and it remains unclear if disparities resolve, persist, or even widen over time. In this study from rural Zimbabwe, we highlight ongoing disparities in cognitive function among CBHF at 7 years of age. Potential biological and environmental explanations for these disparities in CBHF are discussed below. The gap in neurodevelopment of up to 0.28 SD is, if anything, greater than the disparity in neurodevelopmental scores at age 2 years of up to 0.15 SD previously reported in the same cohort [21]. However, a limitation of this comparison is that the measurement at 2 years was across both study districts while at 7 years, this was only in Shurugwi district. Nevertheless, these ongoing modest reductions in cognitive function, many years after exposure to HIV in utero, may have a substantial long-term impact on human capital in areas of high HIV prevalence. There is an evident need to understand the underlying drivers of these differences, so that appropriate interventions can be deployed to improve long-term outcomes in this growing population of children.

The reduction in performance in CBHF was consistent across multiple domains of cognitive function including cognitive processing, academic function, and executive function. These tools had been specifically adapted [42] or developed for children at this age and in this setting [23]. CBHF compared to CHU had a 0.3 standard deviation reduction in the Mental Processing Index, representing the overall cognitive processing score, together with lower school achievement and executive function scores. The consistency across a range of cognitive domains has been previously shown in early life [8]. In a meta-analysis of 11 studies (6 outside the USA including Colombia, Democratic Republic of Congo, South Africa, Tanzania, Uganda, and Zimbabwe), CBHF showed reduced neurodevelopment compared to CHU at young ages, although sample sizes were small [41]. More recent studies have shown increased risk of language delay in CBHF [43,44], but most studies are from children aged below 2 years. Similar decrements in neurodevelopment were found in this cohort at age 2 years, with an effect size of 0.15 SD reduction in the Malawi Developmental Assessment Tool [21]. Here, we show that effects persist across cognitive domains by 7 years of age and may in fact have increased, with a 0.28 SD difference between groups for the main outcome of the Mental Processing Index (MPI) from the Kaufman Assessment Battery for children. Exploratory post hoc analysis provided evidence that reductions occurred across all subtests of these assessments, with the exception of the Flanker test in the executive function test battery (Table E in S1

Text). There was some evidence that CBHF had smaller head circumference at 7 years of age, though this difference was not significant after adjustment for multiple comparisons. This has previously been observed in early life among CBHF [45], and the current cohort had a smaller head circumference at age 18 months (Table C in S1 Text). Reduced postnatal head circumference at age 2 years has been strongly associated with poor neurodevelopmental outcomes [46]. Evidence of structural differences among CBHF is emerging, with a reduction in grey matter volume apparent as early as 3 weeks of age using magnetic resonance imaging [47], or diffuse tensor imaging combined with neuropsychological testing [48]. Head circumference is predominantly a marker of early-life growth, particularly in the first 2 years [49]. Consistent with poorer early-life growth, CBHF in this substudy also demonstrated reduced length-for-age and a higher proportion of underweight at 18 months (Table C in S1 Text). Despite catch-up in other anthropometric measures over time, there is weak evidence that reduced head growth persists at school-age, consistent with the accompanying reductions in neurodevelopment.

This study has several strengths, particularly the use of an extensively piloted toolbox [23] with suitably adapted assessments [25], which provided simultaneous phenotyping of growth, physical, and cognitive function. The cohort has well-characterised longitudinal HIV exposure status, and includes a rich data set including baseline and contemporary maternal, socioeconomic and nurturing factors for use as covariates. This substudy was broadly representative of the whole SHINE cohort since there were no major differences between those included and excluded in the follow-up study. There are also several study limitations, particularly possible survivor bias due to higher mortality in CBHF [7]. There may also be selection bias due to the high number of relocations since the end of the trial, which may be related to household socioeconomic status. Although we collected data during pregnancy on HIV treatment, it was not available for all women, and this substudy was not powered to evaluate the impact of specific antiretroviral regimens on long-term child health outcomes. We had very little missing data for the outcomes we measured (<3%) but where there were missing covariate data, we included a categorical variable for missingness, which may have reduced the variability in the covariate data in adjusted models. Finally, we measured multiple outcomes and this was an exploratory analysis. We therefore applied a Bonferroni correction to account for multiple comparisons, and we only focused our inferences on effects that were consistent and statistically significant using this approach; however, we cannot rule out the possibility of type 1 inflation error.

This study implies that both universal and HIV-specific risk factors contribute to cognitive disparities [9], and the next steps would be to disentangle these contributions. This study showed that the psychosocial and socioeconomic environment was poorer in CBHF, including worse food security, higher adversity scores, caregivers with fewer years of schooling, and higher caregiver depression scores, which may all affect the way caregivers provide nurturing care and how children learn. However, adjusted models including either contemporary or baseline psychosocial variables did not eliminate cognitive differences between groups, although the disparity was attenuated. Antenatal HIV, ART exposure, co-infections, and greater inflammation may drive biological differences in growth and function which are still evident at 7 years [17]. This is consistent with recent findings from the same cohort showing that CBHF have a distinct inflammatory milieu, partly driven by earlier exposure to CMV viraemia [10]. It is plausible this would affect sensitive processes like neurodevelopment and thus explain the long-term effects presented here. However, we cannot exclude residual confounding due to unmeasured socioeconomic and psychosocial variables that may also drive this difference. A combined intervention approach addressing both universal and HIV-specific pathways is likely needed to reduce the cognitive gap between groups, delivered throughout early life and up to school-age.

There was some evidence of reduced cardiovascular fitness in CBHF demonstrated by a reduction in the level obtained in the shuttle run, reflecting a lower VO$_2$max than CHU, although this did not reach the threshold of statistical significance following Bonferroni correction in adjusted models. Other cohorts should explore this further, since the magnitude of reduction in VO$_2$max observed for CBHF is similar to that previously reported for children with anaemia in Kenya [50], and it is biologically plausible that CBHF could have reduced cardiorespiratory capacity. Emerging data suggest effects of HIV exposure on cardiac structure and function in high-income CBHF cohorts [51,52]. Fetal exposure to ART may impair myocardial growth [52]. Zidovudine has been shown to alter fetal cardiac remodelling and to cause mild antenatal dysfunction [53]. Taken together, further detailed studies of cardiac and lung physiology are needed in cohorts of CBHF to better understand whether there is reduced cardiovascular fitness in mid-childhood and the potential mechanisms.

In conclusion, this study of CBHF and CHU in rural Zimbabwe represents one of the few birth cohorts in sub-Saharan Africa to be followed to school-age to ascertain longer-term outcomes. We identified ongoing vulnerabilities among CBHF in multiple domains of cognitive function, which could affect human capital across the life-course. The pressing goal now is to understand the relative contributions of biological and psychosocial factors to these long-term disparities, to inform future interventions, which could potentially include nurturing care and educational provision for children, combined with improved food security and psychosocial support to caregivers [17]. Given the expanding global population of CBHF, there is a need for further long-term follow-up studies to understand how we can ensure that all children survive and thrive across the life-course.

## Supporting information

**S1 Text. Table A. Baseline and early-life characteristics of HIV–positive women and their infants enrolled and not enrolled in the follow-up cohort.** n: number, SD: standard deviation, IQR: Inter-quartile range, LAZ: Length-for-age Z-score, WAZ: Weight-for-age Z-score, WHZ: Weight-for-height Z-score, HCZ: Head circumference for Z-score, MUACZ: Mid-upper arm circumference Z-score, Hb: Haemoglobin, Chi-Sq: Chi -Square from logistic regression and adjusted for clustering. GEE (robust) Generalised estimating equations with robust variance estimation adjusted for clustering, Somers' D comparison of medians using t-distribution adjusted for clustering. *Of the 267 caregivers, data were not available for 2 women at baseline entry to the trial, and for 7 of the not included group. Applying a Bonferroni correction for multiple comparisons, a P-value <0.001 is considered significant. **Table B. Baseline and early-life characteristics of HIV–negative women and their infants enrolled and not enrolled in the follow-up cohort.** n: number, SD: standard deviation, IQR: Inter-quartile range, LAZ: Length-for-age Z-score, WAZ: Weight-for-age Z-score, WHZ: Weight-for-height Z-score, HCZ: Head circumference for Z-score, MUACZ: Mid-upper arm circumference Z-score, Hb: Haemoglobin, Chi-Sq: Chi -Square from logistic regression and adjusted for clustering. GEE (robust) Generalised estimating equations with robust variance estimation adjusted for clustering, Somers' D comparison of medians using t-distribution adjusted for clustering. There were less mothers than children due to some households having twins. *Note that of the 988 HIV–negative mothers recruited into SHINE follow-up, data were not available for 66 mothers at the baseline visit and similarly for the 2,949 HIV–negative mothers not recruited, data were not available for 182 mothers at the baseline visit. **Note that 1,002 children included 2 children who were HIV–positive as their mothers seroconverted during breastfeeding and the children became HIV–positive. These 2 children were excluded from all analyses. Applying a Bonferroni correction for multiple comparisons (40), a P-value <0.001 is

considered significant. **Table C. Baseline household, maternal and child characteristics between children born HIV-free (CBHF) compared with children who were HIV-unexposed (CHU).** Note that household and maternal characteristics were all measured in pregnancy at the baseline visit. Child characteristics were measured at birth for birth outcomes and when the child was 18 months old for the growth outcomes specified. **Table D. Results of subgroup analysis exploring interaction of sex of children with HIV-exposure for SAHARAN toolbox outcomes.** If the p-value was greater than 0.1, the interaction was not considered significant, hence N/A (not applicable) was completed for the difference between boys and girls. **Table E. Secondary outcomes comparing children born HIV-free (CBHF) and children unexposed to HIV (CHU).** Cognitive function included the Kaufman Assessment Battery for Children (KABC-II) with its 8 subtests Atlantis, Story completion, Number recall, Delayed Atlantis, Rover, Triangles, Word Order and Pattern reasoning. Two from each of these 8 subtests were added together to form the 4 cognitive domains of Sequential, Planning, Learning, and Simultaneous domains. The School Achievement Test (SAT) was formed of numeracy, reading, and writing sections. The Plus-EF total was formed of 3 subtests Multi-Source Interference Test (MSIT), Stars and Flowers and Fish Flanker. The Fine motor (FM) test was measured by sequential finger tapping for both dominant and non-dominant hands, using seconds as a unit and hence a higher number represented slower fine motor coordination. The Strength and Difficulties Questionnaire (SDQ) total was measured using 4 subscales of emotional, conduct, hyperactivity, and inattention with higher scores representing more difficulties. In addition the prosocial subscale was separately measured for positive behaviour. The child socioemotional subscore was the total with one question removed on food security. Grip strength (GS) was measured with the highest value for both dominant and non-dominant hands. Standardised scores were included for broad jump (BJ) and the shuttle run test (Run). The total of the standardised scores provided the physical function score. Blood pressure included pulse pressure as the difference between systolic and diastolic, and included values measured after the shuttle run test. Bioimpedance (BIA) measured raw values of reactance and resistance in Ohms. Model 1 is adjusted for trial factors (arm, study nurse, exact child age, calendar month recruited, temperature, sex). Model 2 is adjusted for trial factors from Model 1 and contemporary factors (socioeconomic status, caregiver depression score (EPDS), household food insecurity (HFIAS), household religion, caregiver social support, caregiver gender norms, caregiver age, caregiver education, adversity score, children's books at home). Model 3 is adjusted for trial factors from Model 1 and early-life factors (length for age Z-score (LAZ) at 18 mo, birthweight, maternal baseline depression score (EPDS), household diet, maternal haemoglobin, socioeconomic status, facility birth, gender norms, and maternal years of schooling). *Note that standardised VO2max score was used as this represented the distribution of shuttle run better by adjusting for differences between the number of sub-levels for each stage of the shuttle run test. **Table F. Summary of findings for children born HIV-free (CBHF) compared with children unexposed to HIV (CHU) at 7 years.** This table summarises the associations observed, SD: Standard deviation.
(DOCX)

**S1 Fig. Directed acyclic graph (DAG) used to determine covariates for Model 2: adjustment by contemporary covariates asked in the contemporary questionnaire.** Variables listed from top to bottom: Caregiver edn: Caregiver schooling in number of years, Caregiver age: age of primary caregiver at 7 year visit, 7yr SES: contemporary socioeconomic status (wealth index), adversity: contemporary adversity score, EPDS: contemporary caregiver Edinburgh Postnatal Depression Score, Gender norms: contemporary caregiver gender norm scale, Social support: contemporary caregiver social support scale, Discipline: child discipline scale, Water

insecurity (HWISE): Household water insecurity experiences scale, Food insecurity (HFIAS): Household food insecurity experiences scale, Books: number of children's books at home, Child schooling: Total child schooling in years and months, Hb: child contemporary haemoglobin measured during visit, CPRS: Child parent relationship scale (measure of nurturing), Religion: household religion, mat HIV: Maternal HIV status during pregnancy (the exposure), 7yr Gro & fnc: child growth, cognitive and physical function at 7 years (outcome), DC: Data collector, sex: Child sex, Arm: SHINE trial intervention arm, Date measure: calendar quarter when measurement performed, Age: exact age of child, Ambient temp: Ambient temperature: average temperature during SAHARAN toolbox measurements, Unmeasured Confounders: unmeasured confounders. Adjustment variables for model 2 were arm, Data Collector, age of child, calendar age recruited, temperature, sex, Socioeconomic status, Caregiver depression measure (EPDS), Household food insecurity (HFIAS), Household religion, Caregiver social support, Caregiver gender norms, Caregiver age, Caregiver education, Adversity score, Children's books at home. Colour coding: Green circle: exposure variable, green arrow: causal path, blue circle with I: outcome variable, blue circles: ancestor of outcome variables, red circle: ancestor of exposure and outcome variables, red arrows: biasing path, black arrows: postulated interactions not on the biasing path, grey circle: unobserved (latent) variable i.e., unmeasured confounders.
(TIFF)

**S2 Fig. Directed acyclic graph used to determine covariates for Model 3: adjustment by baseline covariates.** Variables listed from top to bottom: Mat schooling: Maternal schooling in number of years, mat age: maternal age, SES: baseline socioeconomic status (wealth index), employed: whether mother was employed or not, EPDS: Baseline maternal Edinburgh postnatal depression score, Gender norms: baseline maternal gender norm scale, social support: baseline maternal social support scale, CSI: Coping strategies index (measure of food insecurity), HH diet: household dietary score, Mat Hb: Maternal haemoglobin in pregnancy, Mat diet: maternal diet score, Mat anthro: Maternal anthropometry (note height was used in model), Bweight: child birthweight, mat HIV: Maternal HIV status during pregnancy (exposure), 7yr Gro & fnc: child growth, cognitive and physical function at 7 years (outcome), Anthro at 18 months: child anthropometry at 18 months (length-for-age-z-score used in model), religion: household religion, Hb at 18 months: child haemoglobin at 18 months of age. DC: Data collector, sex: Child sex, Arm: SHINE trial intervention arm, Date measure: calendar quarter when measurement performed, Age: exact age of child, Ambient temp: Ambient temperature: average temperature during SAHARAN toolbox measurements, Confounders: unmeasured confounders. Adjustment variables were: Arm, data collector, age of child, calendar age recruited, temperature, Anthropometry at 18mo, Birthweight, maternal depression score (EPDS), household dietary score, maternal haemoglobin in pregnancy, baseline socioeconomic scale, born in facility, gender norms, maternal years of schooling. DC: Data collector, sex: Child sex, Arm: SHINE trial intervention arm, Date measure: calendar quarter when measurement performed, Exact age: exact age of child, ambient temperature: average temperature during SAHARAN toolbox measurements, Confounders: unmeasured confounders, Colour coding: Green circle: exposure variable, green arrow: causal path, blue circle with I: outcome variable, blue circles: ancestor of outcome variables, red circle: ancestor of exposure and outcome variables, red arrows: biasing path, black arrows: postulated interactions not on the biasing path, grey circle: unobserved (latent) variable ie unmeasured confounders.
(TIFF)

**S1 SHINE Protocol. Sanitation Hygiene Infant Nutrition Efficacy Project.**
(DOCX)

**S1 Statistical Analysis Plan. Statistical analysis plan.**
(DOCX)

## Acknowledgments

We thank Mr Peter Maparunga and Stephen Moyo for assistance with logistics; Mrs Phillipa Rambanepasi, Tracy Muzira, and Karen Gwanzura for assistance with finance; drivers Lovemore Chingaona, Lloyd Goremusandu, and Tawanda. We also thank the Zimbabwe Ministry of Health and Child Care, particularly the Provincial and District Health Executives for invaluable support and advice.

## Author Contributions

**Conceptualization:** Joe D Piper, Clever Mazhanga, Gloria Mapako, Tsitsi Mashedze, Dzivaidzo Chidhanguro, Batsirai Mutasa, Handrea Njovo, Chandiwana Nyachowe, Mary Muchekeza, Kuda Mutasa, Virginia Sauramba, Ceri Evans, Melissa J Gladstone, Jonathan C Wells, Elizabeth Allen, Melanie Smuk, Jean H Humphrey, Lisa F Langhaug, Naume V Tavengwa, Robert Ntozini, Andrew J Prendergast.

**Data curation:** Joe D Piper, Clever Mazhanga, Marian Mwapaura, Gloria Mapako, Idah Mapurisa, Tsitsi Mashedze, Eunice Munyama, Maria Kuona, Thombizodwa Mashiri, Kundai Sibanda, Dzidzai Matemavi, Soneni Nyoni, Asinje Saidi, Manasa Mangwende, Dzivaidzo Chidhanguro, Eddington Mpofu, Joice Tome, Gabriel Mbewe, Batsirai Mutasa, Ceri Evans, Melissa J Gladstone, Jonathan C Wells, Elizabeth Allen, Melanie Smuk, Lisa F Langhaug, Robert Ntozini, Andrew J Prendergast.

**Formal analysis:** Joe D Piper, Marian Mwapaura, Batsirai Mutasa, Bernard Chasekwa, Ceri Evans, Elizabeth Allen, Melanie Smuk, Robert Ntozini, Andrew J Prendergast.

**Funding acquisition:** Joe D Piper, Robert Ntozini, Andrew J Prendergast.

**Investigation:** Joe D Piper, Clever Mazhanga, Marian Mwapaura, Gloria Mapako, Idah Mapurisa, Tsitsi Mashedze, Eunice Munyama, Maria Kuona, Thombizodwa Mashiri, Kundai Sibanda, Dzidzai Matemavi, Monica Tichagwa, Soneni Nyoni, Asinje Saidi, Manasa Mangwende, Dzivaidzo Chidhanguro, Chandiwana Nyachowe, Melissa J Gladstone, Jonathan C Wells, Jean H Humphrey, Robert Ntozini, Andrew J Prendergast.

**Methodology:** Joe D Piper, Clever Mazhanga, Marian Mwapaura, Idah Mapurisa, Tsitsi Mashedze, Eunice Munyama, Maria Kuona, Thombizodwa Mashiri, Kundai Sibanda, Dzidzai Matemavi, Monica Tichagwa, Soneni Nyoni, Asinje Saidi, Manasa Mangwende, Dzivaidzo Chidhanguro, Chandiwana Nyachowe, Kuda Mutasa, Virginia Sauramba, Ceri Evans, Melissa J Gladstone, Jonathan C Wells, Jean H Humphrey, Lisa F Langhaug, Naume V Tavengwa, Andrew J Prendergast.

**Project administration:** Joe D Piper, Clever Mazhanga, Marian Mwapaura, Tsitsi Mashedze, Dzivaidzo Chidhanguro, Handrea Njovo, Chandiwana Nyachowe, Lisa F Langhaug, Robert Ntozini, Andrew J Prendergast.

**Resources:** Eddington Mpofu, Robert Ntozini, Andrew J Prendergast.

**Software:** Marian Mwapaura, Eddington Mpofu, Joice Tome, Gabriel Mbewe, Batsirai Mutasa.

**Supervision:** Joe D Piper, Clever Mazhanga, Marian Mwapaura, Dzivaidzo Chidhanguro, Joice Tome, Batsirai Mutasa, Mary Muchekeza, Virginia Sauramba, Melissa J Gladstone,

Jonathan C Wells, Elizabeth Allen, Melanie Smuk, Lisa F Langhaug, Naume V Tavengwa, Robert Ntozini, Andrew J Prendergast.

**Validation:** Joe D Piper, Clever Mazhanga, Marian Mwapaura, Ceri Evans, Melanie Smuk, Robert Ntozini, Andrew J Prendergast.

**Visualization:** Joe D Piper, Joice Tome, Bernard Chasekwa, Lisa F Langhaug, Robert Ntozini, Andrew J Prendergast.

**Writing – original draft:** Joe D Piper, Andrew J Prendergast.

**Writing – review & editing:** Joe D Piper, Clever Mazhanga, Marian Mwapaura, Gloria Mapako, Idah Mapurisa, Tsitsi Mashedze, Eunice Munyama, Maria Kuona, Thombizodwa Mashiri, Kundai Sibanda, Dzidzai Matemavi, Monica Tichagwa, Soneni Nyoni, Asinje Saidi, Manasa Mangwende, Dzivaidzo Chidhanguro, Eddington Mpofu, Joice Tome, Gabriel Mbewe, Bernard Chasekwa, Handrea Njovo, Chandiwana Nyachowe, Mary Muchekeza, Kuda Mutasa, Virginia Sauramba, Ceri Evans, Melissa J Gladstone, Jonathan C Wells, Elizabeth Allen, Melanie Smuk, Jean H Humphrey, Lisa F Langhaug, Naume V Tavengwa, Robert Ntozini, Andrew J Prendergast.

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
