## [Editor Report · Decision Letter 0]

3 Jan 2024

Dear Dr Piper, 

Thank you for submitting your manuscript entitled "Growth, physical and cognitive function in children who are born HIV-free: school-age follow-up of a cluster-randomized trial in rural Zimbabwe" for consideration by PLOS Medicine.

Your manuscript has now been evaluated by the PLOS Medicine editorial staff and I am writing to let you know that we would like to send your submission out for external peer review.

Please re-submit your manuscript within two working days, i.e. by Jan 05 2024 11:59PM.

Kind regards,

Alexandra Schaefer, PhD

Associate Editor

PLOS Medicine

---

## [Decision Letter · Decision Letter 1]

13 Feb 2024

Dear Dr. Piper,

Thank you very much for submitting your manuscript "Growth, physical and cognitive function in children who are born HIV-free: school-age follow-up of a cluster-randomized trial in rural Zimbabwe" (PMEDICINE-D-23-03875R1) for consideration at PLOS Medicine. 

Your paper was evaluated by an associate editor and discussed among all the editors here. It was also discussed with an academic editor with relevant expertise, and sent to independent reviewers, including a statistical reviewer. The reviews are appended at the bottom of this email and any accompanying reviewer attachments can be seen via the link below:

[LINK]

In light of these reviews, I am afraid that we will not be able to accept the manuscript for publication in the journal in its current form, but we would like to consider a revised version that addresses the reviewers' and editors' comments. Obviously we cannot make any decision about publication until we have seen the revised manuscript and your response, and we plan to seek re-review by one or more of the reviewers. 

Please use the following link to submit the revised manuscript: https://www.editorialmanager.com/pmedicine/

We expect to receive your revised manuscript by Mar 05 2024. However, if this deadline is not feasible, please contact me by email, and we can discuss a suitable alternative.

Don't hesitate to contact me directly with any questions (aschaefer@plos.org). If you reply directly to this message, please be sure to 'Reply All' so your message comes directly to my inbox.

We look forward to receiving your revised manuscript. 

Sincerely,

Alexandra Schaefer, PhD

PLOS Medicine

plosmedicine.org

ACADEMIC EDITOR COMMENTS

It is a very relevant, well-expected study. The following key issues need to be attended to specifically:

1) From Reviewer 1: Please address the reviewer's concern about the imbalance in the reporting of results (unadjusted versus adjusted "p values" - rather than odds ratios). Please tone down the claims, as the differences are real but do not involve an overwhelming number of outcome variables.

2) From Reviewer 5: Please address the potential lack of representativeness of the sample due to loss to follow-up, etc. Also address related issues such as missing values. These have been clearly outlined by Reviewer #5 and need to be addressed. 

3) From Reviewer 3 (statistical reviewer): In addition to the issues raised by Reviewer #1, the author needs to pay attention to the statistical issues raised by Reviewer #3. They are not major, but need to be addressed to clean up the paper.

GENERAL COMMENTS

1) Please include page numbers and line numbers in the manuscript file. Use continuous line numbers (do not restart the numbering on each page). For review purposes, we started counting the Abstract as page 1.

2) Please include the study protocol document and analysis plan, with any amendments, as Supporting Information to be published with the manuscript if accepted.

3) Please report your study according to CONSORT, but explicitly state that it is a sub-study. Please ensure that the Abstract describes the main points of the original trial (2-3 sentences, including study population, study dates, intervention and primary outcome), but that the main part of the Abstract describes the details of the follow-up study. 

4) Please cite your Supporting Information as outlined here: https://journals.plos.org/plosmedicine/s/supporting-information

COMPETING INTEREST

All authors must declare their relevant competing interests per the PLOS policy, which can be seen here:

https://journals.plos.org/plosmedicine/s/competing-interests

For authors with ties to industry, please indicate whether any of the interests has a financial stake in the results of the current study.

DATA AVAILABILITY 

PLOS Medicine requires that the de-identified data underlying the specific results in a published article be made available, without restrictions on access, in a public repository or as Supporting Information at the time of article publication, provided it is legal and ethical to do so. Please see the policy at http://journals.plos.org/plosmedicine/s/data-availability and FAQs at http://journals.plos.org/plosmedicine/s/data-availability#loc-faqs-for-data-policy

The Data Availability Statement (DAS) requires revision. For each data source used in your study: 

ABSTRACT

1) PLOS Medicine requests that main results are quantified with 95% CIs as well as p values. When reporting p values please report as p<0.001 and where higher as the exact p value p=0.002, for example. For the purposes of transparent data reporting, if not including the aforementioned please clearly state the reasons why not.

2) Throughout, suggest reporting statistical information as follows to improve clarity for the reader “22% (95% CI [13%,28%]; p</=)”. Please amend throughout the abstract and main manuscript. Please note the use of commas to separate upper and lower bounds, as opposed to hyphens as these can be confused with reporting of negative values.

3) When a p value is given, please specify the statistical test used to determine it. 

4) Please ensure that all numbers presented in the abstract are present and identical to numbers presented in the main manuscript text.

5) If not already included, please include the study design, population and setting, number of participants, years during which the study took place, length of follow up, and main outcome measures.

6) Please include the important dependent variables that are adjusted for in the analyses.

7) Please define all abbreviations including those for statistical reporting at first use.

AUTHOR SUMMARY

At this stage, we ask that you include a short, non-technical Author Summary of your research to make findings accessible to a wide audience that includes both scientists and non-scientists. The authors summary should consist of 2-3 succinct bullet points under each of the following headings:

• Why Was This Study Done? Authors should reflect on what was known about the topic before the research was published and why the research was needed.

• What Did the Researchers Do and Find? Authors should briefly describe the study design that was used and the study’s major findings. Do include the headline numbers from the study, such as the sample size and key findings.

• What Do These Findings Mean? Authors should reflect on the new knowledge generated by the research and the implications for practice, research, policy, or public health. Authors should also consider how the interpretation of the study’s findings may be affected by the study limitations. In the final bullet point of ‘What Do These Findings Mean?’, please describe the main limitations of the study in non-technical language.

Author Summary should immediately follow the Abstract in your revised manuscript. This text is subject to editorial change and should be distinct from the scientific abstract. Please see our author guidelines for more information: https://journals.plos.org/plosmedicine/s/revising-your-manuscript#loc-author-summary

INTRODUCTION

1) If there has been a systematic review of the evidence related to your study (or you have conducted one), please refer to and reference that review and indicate whether it supports the need for your study.

2) Please ensure to define abbreviations at first use, such as “IQ”. Please revise throughout the main manuscript.

METHODS AND RESULTS

1) PLOS Medicine requests that main results are quantified with 95% CIs as well as p values. When reporting p values please report as p<0.001 and where higher as the exact p value p=0.002, for example. For the purposes of transparent data reporting, if not including the aforementioned please clearly state the reasons why not. We suggest reporting statistical information as detailed above – see under ABSTRACT

2) Please present numerators and denominators for percentages, at least in the Tables [not necessarily each time they're mentioned].

3) Please define HUU at first use.

DISCUSSION

Please present and organize the Discussion as follows: a short, clear summary of the article's findings; what the study adds to existing research and where and why the results may differ from previous research; strengths and limitations of the study; implications and next steps for research, clinical practice, and/or public policy; one-paragraph conclusion.

FIGURES

For all Figures, please ensure that you have complied with our figures requirements http://journals.plos.org/plosmedicine/s/figures.

1) Please provide titles and legends for all figures (including those in Supporting Information files).

2) Please in the figure legend/description, define abbreviations used in each figure (including those in Supporting Information files).

3) Figure 1: Please use “selected” instead of “randomly selected” (“1349 Randomly selected”, third box on the main CHU path).

4) Figure 0-1/0-2: Please define the meaning of the colors used in the graphs.

TABLES

1) Please note the use of commas to separate upper and lower bounds, as opposed to hyphens as these can be confused with reporting of negative values. Suggest reporting statistical information as detailed above – see under ABSTRACT

2) Please provide titles and legends for all tables (including those in Supporting Information files).

3) Please define all abbreviations used in the table below each table (including those in Supporting Information files). For example, for table one: SOC, IYCF, WASH, SES, SD, CBHF, CHU.

SUPPLEMENTARY MATERIAL

1) For supplementary figures and tables, please see the general comments under TABLES and FIGURES and amend accordingly.

2) We suggest reporting statistical information as detailed above – see under ABSTRACT. Please define all numerical values.

3) As for the main manuscript, please indicate whether analyses are adjusted to help facilitate transparent data reporting please also detail the factors adjusted for and present the unadjusted analyses for comparison. If not, please clearly state the reasons why not.

4) Please revise the references according to the comments below - see under REFERENCES.

REFERENCES

1) PLOS uses the numbered citation (citation-sequence) method and first six authors, et al.

2) Please ensure that journal name abbreviations match those found in the National Center for Biotechnology Information (NCBI) databases (http://www.ncbi.nlm.nih.gov/nlmcatalog/journals), and are appropriately formatted and capitalised.

3) Where website addresses are cited, please specify the date of access (e.g. [accessed: 12/06/2023]).

4) Please also see https://journals.plos.org/plosmedicine/s/submission-guidelines#loc-references for further details on reference formatting. 

Comments from the reviewers:

Reviewer #1: Main Comments

This study uses follow-up data collected among former SHINE trial participants in Zimbabwe to compare children exposed to HIV in utero to non-exposed children at age 7. The study is very carefully executed, and the set of measures applied to assess various domains of development and physical performance is truly impressive - congratulations to the study team for putting this battery together and for making all of this happen in the field!

I think this is a very strong and interesting paper. My only major concern is the rather striking imbalance in the reporting of results - I honestly have not seen such one-sided reporting of study results in quite a while. Reading the abstract and the conclusions, one might get the impression that exposed children perform consistently worse on all domains than unexposed children. If you look at the tables, this is really not the case. To start out with, the paper reports in total 27 hypotheses tests (if I counted correctly) across Tables 2-4. Assuming that the zeros in parentheses reported (on a side note: adding one more digit after the comma would really help) mean that the CIs cross zero, I counted 5/27 estimates with a p-values < 0.05, which is not nearly as clear a pattern as the authors suggest. If any kind of multiple testing correction would be applied here (Bonferroni or similar), it is not clear if any of these estimates would reach "statistical significance". 

I would recommend that the authors first add both unadjusted and corrected p-values for all estimates in Tables 2-4, and then carefully revise the abstract and Discussion sections in light of the not-so-overwhelming evidence. Simply reporting results that are aligned with the hypothesis of the authors is really not okay, and seriously undermines the scientific validity and quality of this otherwise so carefully done study. 

Minor Comments

- Abstract Methods & Findings: given the small proportion of kids selected for this study, it would be good to report attrition rates (% of targeted kids assessed) 

- Abstract: mean difference was lower - should either be mean difference was -3, or the mean score was 3 points lower. Differences in points are not very informative if the scale is not presented - I would suggest to either report normalized z-score differences here or define the mean and SD of scales used.

- Page 6: the pre-analysis plan looks very different to what is presented here - is there a separate paper looking at long term treatment effects? Please explain how this paper is related to the study described in the protocol - is this just covering hypothesis 3d? The way this is presented here and on page 9 (statistical analysis) one may think that there is a plan for this paper specifically which is quite misleading.

- Page 9: please clarify what measure of socioeconomic status you used: given the strong links between poverty and HIV carefully controlling for differences in incomes and/or asset holdings seems important here.

- Figure 1: I think it would be useful to also report % of targeted populations here - if this is too much for the figure, the % of children assessed should be reported in the text (e.g. 264/421 = 63%).

- Page 14: .."although both were relatively high" - not sure what that means - would cut this comment.

- Page 14: higher depression scores - verb is missing.

- Table 1: I was surprised by 3 years of schooling at age 7 - what kinds of schooling is counted/reported here?

- EDPS: would be better to report % in critical ranges here rather than mean scores.

- Page 15: please avoid statements like "still strong evidence of difference" - this does not have any real meaning - provide mean differences and 95% CIs instead.

- Table 2: with zero digits after the comma it is hard to say if the CIs cross zero here - is this a journal requirement? If not, I would show at least one digit after the comma. Overall reporting seems also a bit biased - out of six outcomes analyzed, you find differences only for 3 - this should also be clearer in the abstract and conclusions. The "taken together" summary is really not appropriate on page 16 - see also my main comment above.

Reviewer #2: General Comments: This is a timely and comprehensive study among a well characterized group of Zimbabwean children who were perinatally HIV exposed but uninfected; and presents growth and neurodevelopmental follow up data in their early school years.. The children and their caretakers took part in the SHINE (Sanitation Hygiene Infant Nutrition Efficacy) cluster randomized trial in rural Zimbabwe. Initial assessment of SHINE interventions and growth/developmental outcomes were conducted through age 2 years; and revealed some early differences in growth and early development comparing the HIV/ARV exposed uninfected group and children of similar age, and sex who were born to HIV negative women. This study traces and follows up later developmental and growth outcomes at age 7 years, again comparing physical and neurodevelopmental outcomes of HIV/ART perinatally exposed uninfected children to outcomes among HIV unexposed children both to HIV negative mothers in Zimbabwe. The findings are of interest and raise questions about potential longer term negative effects of perinatal HIV and/or antenatal ART exposure both for neurodevelopment and physical outcomes among HIV exposed uninfected children. However given that there are also differences in socioeconomic and maternal education risk factors between the two groups, it is not possible to tease out the specific impact of either perinatal HIV/ART exposure on the described outcomes.

Specific Comments:

Background/ Introduction: Well written and covers current literature which have noted increased mortality and morbidity including poor growth and neurodevelopmental outcomes for HIV/ART exposed uninfected children compared to HIV unexposed children. 

Methods: 

Well described as to the specific school and neurocognitive testing, physical functioning tests and growth measures performed. The tests appear generally culturally appropriate. 

Statistical analyses approach appears appropriate including the 3 models presented. 

Results: 

Growth, neuropsychological testing results and physical function test results for the HIV exposed uninfected and the HIV unexposed children originally in the SHINE trial at age 7 years, are well summarized in the text and the tables of the 3 models and their results are also clear. Of interest is some reduction in physical functioning performance. The results also do indicate some significant differences by group in well recognized maternal education, depression and sociodemographic factors that may affect children's growth and developmental outcomes. Of note, the investigators found lower cognitive performance for HIV/ART perinatally exposed but HIV negative children compared to HIV unexposed children at age 7 years. In addition, with some differences noted by sex, the HIV/ART perinatally exposed children demonstrated reduction in measures of physical performance functioning for several measures. 

Discussion: Highlights the major findings from the analyses, with comparisons to findings other studies. The authors also acknowledges both strengths and weaknesses of the study findings. 

In sum, this follow up study of Zimbabwean children previously enrolled in the SHINE trial is well written and has a number of strengths. The findings are important in documenting longer term follow up of physical growth and neurodevelopmental outcomes during the early school age years for HIV/ART perinatally exposed children living in Zimbabwe, a resource limited international setting; and is strengthened with the ability to compare their outcomes to those of HIV unexposed Zimbabwean children of similar age and sex. 

Reviewer #3: Statistical review

This paper reports an observational study from within a cluster randomised trial that compares school-aged outcomes between non-HIV infected children who were exposed to HIV during pregnancy and those who weren't exposed to HIV. 

The authors show there are significant differences between groups in these outcomes, although whether this is due to the actual exposure itself or associated factors (such as the mother being HIV-positive) is not possible to conclude from this data. 

I had some comments on the statistical methods and reporting, which are provided below.

1. Title: I felt the title implied that the results in this paper would be between randomized groups, so recommend modifying to something like 'an observational sub-study of a cluster randomized trial' or suchlike.

2. Abstract "We found no evidence of differences in other growth" - I'd add 'significant' to this.

3. Page 6 - when the protocol/SAP for the follow-up study are referred to, it might be useful to add that this paper represents Objective 3d) of the follow-up study.

4. Page 9 - when the authors refer to handling within-cluster correlation, I am not sure whether it is sufficiently clear in this paper what clusters are. The only previous reference to 'cluster' is that the SHINE was cluster-randomized.

5. Page 9 - I appreciate that DAGs were used to identify variables to adjust for, but having a bit more information on how the DAGs led to the chosen variables would be useful here. Was the purpose to avoid adjusting for mediators and colliders? Personally I was not sure whether it is wise to adjust for variables measured post-birth when looking at the association between a pre-birth exposure and an outcome so it would be good if more could be provided in the paper's methods section about that type of thing. Perhaps some additional narrative in the supplementary material alongside the DAGs would help. 

6. Page 20 - for the subgroup analyses I would add the p-value for the interaction test when they are mentioned. At the end of the paragraph, I would recommend mentioning the number of 'other growth or body composition outcomes' that were not significant (I count 23). Overall I don't think 4 out of 27 significant results at p<0.1 is beyond what one would expect by chance here, so making these changes should hopefully ensure the results are not overinterpreted.

James Wason

Reviewer #4: This study is commendably well-conducted, particularly regarding the comprehensive range of measurements and the meticulous application of rigorous instruments. My overall critique, however, hinges on the pivotal nature of the research question, which, in my opinion, could be further refined by leaning into the biological hypothesis more clearly. I recognize that the evidence to date is far from strong, but that is why this paper becomes more interesting if it does so.

One important issue in studies on how HIV exposure impacts child development is related to whether the associations are due to socio-economic or biological factors. From my perspective, the socio-economic hypothesis is both less compelling and less plausible. It is unlikely that HIV exposure alone significantly alters a child's social standing or existence to an extent that affects developmental outcomes. Instead, socio-economic links to development are more likely and highly plausibly confounded by factors associated with a mother's HIV status (and therefore her child's consequent exposure), and poor development. Despite efforts to measure various factors, many studies do not capture the complexity of individual circumstances or the socio-economic nuances of the mother. Thus, residual confounding or unmeasured confounding is the most probable explanation for the observed association between HIV exposure without infection and negative outcomes in children.

On the other hand, a biological hypothesis is more interesting. There are documented potential mechanistic effects of HIV exposure on development, as noted in the discussion. For example, the discussion of medication effects on HIV-exposed children and hypotheses around inflammation opens the door to more biologically centered hypotheses. By embracing a biological hypothesis, the evidence here could become more open to falsification but at the same time potentially more compelling. Imagine if the hypothesis was that HIV exposure itself has developmental effects that cannot be easily explained by social and psychological factors, and therefore support clinical effects mediated by plausible biological mechanism. 

Clarification is needed regarding the SHINE trial's recruitment and enrollment procedures. The significance of these findings largely depends on their capacity to represent population-level effects of HIV exposure. Selection bias could profoundly influence the observed associations. Although the follow-up observational study appears to have equitably enrolled HIV-exposed and unexposed participants, the initial trial enrollment's potential biases have not been sufficiently detailed.

An important oversight from the report is the description of the clinical significance of the observed differences. With diverse tests assessing neurocognitive and cardiovascular functions, the relevance of these disparities is unclear. Without understanding their place on a standard distribution curve, it's difficult to assess the impact of these differences on the children's life spans and regional public health. I suspect they are large, but I am not expert in any one of these and so I am uncertain. 

The paper's long follow-up period is a key asset, recognizing that HIV-exposed yet uninfected children face a range of adverse outcomes, including increased mortality and developmental challenges.

The use of directed acyclic graphs stands out methodologically, yet the rationale behind the selection of variable groupings requires more explanation to enhance comprehension. The justification for employing three different models in the study remains somewhat opaque. While I presume all variable sets are sufficient to block all backdoor paths, the purpose of the additional adjustment sets, whether for efficiency or to address uncertainties around the minimal variable set needed to close all confounding pathways, or other rationale, is not immediately apparent.

Other strengths of the paper include the impressive stability of differences after adjustment and the novel exploration of child outcomes beyond two years, a period not extensively covered in existing research. This leaves open questions about whether these disparities will resolve, persist, or widen as time progresses.

To me, the issue of multiple hypothesis testing does not diminish the study's credibility. 

Reviewer #5: Thank you for the opportunity to review this manuscript. I would like to congratulate the authors on a well-executed study and clearly written manuscript. The paper examines the long-term differences in cognitive and growth outcomes between children born HIV-free and children HIV unexposed who participated in the SHINE trial. Overall, the results are interesting and fill an important gap in the literature. I have laid out a few major and some minor issues below that can help improve the paper. 

MAJOR comments 

* Representativeness of the sample - the authors argue that the analytical sample is likely representative of the SHINE cohort (p23), but in my opinion they fail to show this clearly. 

o First, they only selected one district for the follow up and it's not clear why only one district was selected or how it was selected. Were there any major differences between the two districts at the start of the trial? 

o Second, sup tables 1a and 1b compare those included in the 7-year follow-up to those excluded at baseline. Many of the differences are not only significant but also quite large in terms of magnitude. In addition, the Ns reported don't match the flow chart, e.g., table s1a 267 included in follow up when flow says 264; 459 not included when flow says 421 live births. These Ns need to be reconciled

o Third, the sample at 7y is not compared to the sample at 18 mo. Some kids were lost to follow up during the trial and others afterwards. Is the sample at 7 y representative of the cohort that completed the trial? 

* Persistent effects of HIV exposure - the authors can make a stronger argument that the "effects" of HIV exposure persist later in life by showing what the effects were at 18 months of age. You've referenced a prior paper but the sample is different there. And as I argue in my previous point, I'm not convinced by what is shown in the current paper that the subsample followed at 7 y is representative of the full SHINE cohort. It would strengthen the paper to show results at 18 mo for the sample being analyzed at 7 y. 

* Related, you note that that the disparity between CBHF and CHU is in fact larger than at 2 years of age, but these results aren't shown or formally compared in the text. I think this finding needs to be highlighted a bit more as it has important programmatic implications. 

* Attrition - overall, attrition is pretty high which is not unexpected given the long follow up. Of the 376 CBHF who completed SHINE, you enrolled 264, which is ~70%. Did you conduct any sensitivity analyses (eg IPW) to account for attrition? 

* Child assessments - a bit more information in the main text would be useful, particularly around the validation/adaption of the tools (you don't note until the discussion that there was an article on developing the toolbox) and training received by the nurses. It would also be helpful to describe the order in which the assessments were administered (was it always the same?) and whether they were administered at the same time of the day. These and other administration factors can influence how children perform on the assessments. Although you do control for nurse in the regressions, you don't control for other factors that can influence performance at the time of assessment. 

* Missing data - there is no mention of how missing data on any of the variables (outcome or covariates) were handled. Some of the table notes mention missing data, but this should be consistently described in the methods.

* Objective of the three tested models - it would be good to specify the objective of examining the three different models. Is to understand whether contremporary or baseline factors matter more? If so, there should be some discussion to this effect in the Discussion. As it stands, there is little value in showing all three models in the main text. 

MINOR Comments 

* The supplement is a little messy. Tables have multiple titles/notes. Not all are referenced in the main text. Main text presents variables that aren't in the supplemental tables. This made it a little difficult to follow. 

* Some of the test from which p-values are derived for the tables are not explicitly described in the methods. 

* Table 1 - these are the "contemporary" characteristics, right? Given that your data come from a trial, calling them baseline characteristics, implies you showing characteristics at the time of enrolment into the trial, which isn't the case. 

* The Discussion largely reports on whether findings are consistent or not with prior studies. There is very little on programme and research implications, and it can benefit from a few sentences on what these findings mean for programs in Zimbabwe and other LMICs. 

* There are lot of assessments (which is very impressive!) and indicators, with some information in the main text and some in several parts of the supplement. This makes it a little challenging for readers who aren't familiar with all the assessments and scores derived from them. Although not necessary per se, I think the paper can benefit from presenting all this information in a table format, particularly for the cognitive and growth outcomes presented in the main text. 

* Subgroup analysis - did the interventions modify any of the differences between CBHF and CHU? I assume not since the data were not presented in the text or supplemental table. It would be good to include explicitly since such results have implications for whether or not early life interventions can help reduce disparities between CBHF and CHU later in life.

[LINK]

---

## [Decision Letter · Decision Letter 2]

24 Jun 2024

Dear Dr. Piper,

Thank you very much for re-submitting your manuscript "Growth, physical and cognitive function in children who are born HIV-free: school-age follow-up of a cluster-randomized trial in rural Zimbabwe" (PMEDICINE-D-23-03875R2) for review by PLOS Medicine.

Thank you for your detailed response to the editors' and reviewers' comments. I have discussed the paper with my colleagues and the academic editor, and it has also been seen again by four of the original reviewers. The changes made to the paper were mostly satisfactory to the reviewers. As such, we intend to accept the paper for publication, pending your attention to the remaining reviewer and editorial comments below in a further revision. When submitting your revised paper, please once again include a detailed point-by-point response to the editorial comments.

[LINK]

We ask that you submit your revision within 1 week (Jul 01 2024). However, if this deadline is not feasible, please contact me by email, and we can discuss a suitable alternative.

Please do not hesitate to contact me directly with any questions (atosun@plos.org). If you reply directly to this message, please be sure to 'Reply All' so your message comes directly to my inbox.

We look forward to receiving the revised manuscript.   

Sincerely,

Alexandra Tosun, PhD

Associate Editor 

PLOS Medicine

plosmedicine.org

Requests from Editors:

DATA AVAILABILITY STATEMENT

Please update the statement in the online submission form with the details provided in ll.273-278 and remove the data availability statement from the main text.

ABSTRACT

1) ll.3-4: “Globally, over 16 million children were exposed to HIV during pregnancy but remain HIV free at birth and throughout childhood.” – please add the time frame/point.

2) l.32, please change to: “…number of children’s books)…”

3) l.38: Please write ‘SD’ and ‘CI’ in full.

4) l.45, please change to: “In this study, we found that … had …”.

AUTHOR SUMMARY

1) l.52: “Over 16 million children globally were born HIV-free (CBHF) to mothers living with HIV.” – please add the time frame/point.

2) l.59: Please write ‘SHINE’ in full.

3) In the final bullet point of the ‘What do these findings mean’ sub-section, please include the main limitations of the study in non-technical language.

INTRODUCTION

l.92: Please define ‘CMV’ at first use.

RESULTS

1) l.282: Please define ‘IQR’ at first use.

2) Figure 1: Please define ‘CBHF’, ‘CHU’, ‘SHINE’ and ‘SFU’.

3) ll.307-308, please change to: “CBHF caregivers also had higher depression scores (4.1 vs 3.2) compared to CHU.”

4) l.315: Please define ‘SD’ at first use.

5) Table 1: Please define ‘CBHF’, ‘CHU’, ‘SES’, ‘SHINE’ (at first use).

6) ll.321-322, please change to: “Baseline factors measured during participation in the original trial are shown in Tables S1 and S2 [23].”

7) ll.323-323: “All scales are explained in supplementary materials.” – please specify.

8) l.334: Please add statistical information, e.g. “(mean (SD) -0.46 [1.07] vs -0.21 [0.98])”.

9) Table 2: Please define ‘CI’, ‘CBHF’, ‘CHU’, ‘EF’, ‘SDQ’ (or ensure to introduce the abbreviations properly).

10) l.356/367/381: Please change to ‘at 18 months’.

11) Table 3: Please define ‘CI’, ‘CBHF’, ‘CHU’.

12) Table 4: Please define ‘CI’, ‘CBHF’, ‘CHU’, ‘MUAC’.

13) l.400, please change to: 'sex of children'. Please also revise Table S4 accordingly.

14) l.405/406: “CHU boys/CBHF boys”” – please re-word and use patient-centered language, e.g. “boys in the CHU group”. Please revise accordingly through the main text.

DISCUSSION

1) ll.421-423: “The gap in neurodevelopment of up to 0.28 SD is, if anything, greater than the disparity in neurodevelopmental scores at age 2 years of up to 0.15 SD previously reported in the same cohort [21]. – The comparison, while interesting, has its limitations due to different study cohorts being compared, which may be worth clarifying and reiterating at this point.

2) l.436: "(6 outside the USA)" - Given that the current study was conducted in Zimbabwe, the information "6 outside the USA" seems less helpful. Could this be further specified?

3) l.443: Please ensure to introduce abbreviations, such as ‘MPI’, or write in full.

4) l.452: Please write ‘MRI’ in full.

REFERENCES

1) Where website addresses are cited, please specify the date of access using use the word “accessed” (e.g. [accessed: 10/04/2024]).

2) Please ensure that journal name abbreviations match those found in the National Center for Biotechnology Information (NCBI) databases (http://www.ncbi.nlm.nih.gov/nlmcatalog/journals), and are appropriately formatted and capitalised. For example, “Frontiers in Pediatrics” in reference [1] should be “Front Pediatr.

3) RE reference [25, 26] – it seems that the two references are identical. Please check and update the status (i.e., “[version 2; peer review: 2 approved]”).

4) Please revise the supporting information references according to the comments above.

SUPPLEMENTARY MATERIAL

1) Please revise the supporting information, especially the definition of abbreviations.

2) S2 Figure: Please note that in the file you submitted (and in the file ‘CBHF_CHU_PlosMed_Supp_info.docx’), the graph appears to be cut off on the left and right sides.

3) In the published article, supporting information files are accessed only through a hyperlink attached to the captions. For this reason, you must list captions at the end of your manuscript file. You may include a caption within the supporting information file itself, as long as that caption is also provided in the manuscript file. Do not submit a separate caption file.

When SI files are contained with a single file:

Please label the file as ‘S1 Supporting Information’.

Please apply alphabetical labelling to each table and figure contained within the S1 file. For example, ‘Fig A’ to ‘Fig Z’ and ‘Table A’ to ‘Table Z’.

Plain text does not need to be labelled and can just be given a title as necessary. For example, ‘Statistical Analysis Plan’.

Please cite tables/figures as ‘Fig A in S1 Supporting Information’ and/or ‘Table A in S1 Supporting Information’, for example.

Please cite plain text as, ‘Statistical Analysis Plan in S1 Supporting Information’, for example.

When SI files are uploaded as separate files:

Please label tables as ‘S1 Table’ (so on) and figures as ‘S1 Fig’ (and so on).

Any additional documents (protocols/analysis plans etc.) can be labelled as ‘S1 Protocol’, for example. Please cite items as exactly as labelled.

SOCIAL MEDIA

To help us extend the reach of your research, please provide any X (formerly known as Twitter) handle(s) that would be appropriate to tag, including your own, your co-authors’, your institution, funder, or lab. Please enter in the submission form any handles you wish to be included when we post about this paper.

Comments from Reviewers:

Reviewer #1: I would like to thank the authors for the careful revision of the manuscript - I think the draft is much improved and much easier to follow now. I am still not fully happy with the presentation of results though:

- Figure 1: it would be nice to also have the percentages included in each step here. I think reporting the 990 CHU as «49%» is quite misleading, considering that you selected only some children for this follow-up - I think it would be more appropriate to report this as percentage of the 1349 selected shown in the figure. If this is the correct denominator, follow-up rates would be slightly higher in the CHU group, but these differences are likely not statistically significant (would be good to test and report this).

- Methods: some more details on the multiple testing corrections would be helpful here - how did you determine the 0.002 cutoff? Was this based on the 27 comparisons mentioned in the legend of the table? If yes, the correct cutoff would be 0.05/27, which would be around 0.00185. Please provide a rationale for and more detail on this in the Methods section. Given that hypotheses are likely not independent here, other types of corrections (step-down, or FDR) would likely be a bit more forgiving here.

- Tables 2-4 claim that you corrected for multiple testing, and yet, the table shows only uncorrected p-values. Please kindly indicate differences that are below this threshold with bold font or some other markers.

- Abstract: you should state clearly here that 27 variables were analyzed, and that statistically different results were only found for 2 (or 3?) of these variables. Is the EF p-value below or above this cutoff?

Reviewer #3: A detailed response to my comments was not provided. However, looking over the changes made it appears all my previous comments have been addressed. I have no further issues to raise.

Reviewer #4: I am satisfied with the answers the authors have provided. 

Reviewer #5: The authors have adequately addressed all my comments. I have no further comments on the paper.

[LINK]

General Editorial Requests

---

## [Editor Report · Decision Letter 3]

26 Aug 2024

Dear Dr Piper, 

On behalf of my colleagues and the Academic Editor, James K Tumwine, I am pleased to inform you that we have agreed to publish your manuscript "Growth, physical and cognitive function in children who are born HIV-free: school-age follow-up of a cluster-randomized trial in rural Zimbabwe" (PMEDICINE-D-23-03875R3) in PLOS Medicine.

I appreciate your thorough responses to the reviewers' and editors' comments throughout the editorial process. We look forward to publishing your manuscript, and editorially there are only a few remaining minor stylistic/presentation points that should be addressed prior to publication. We will carefully check whether the changes have been made. If you have any questions or concerns regarding these final requests, please feel free to contact me at atosun@plos.org.

Please see below the minor points that we request you respond to:

1) Abstract: When reporting p-values, please use either lowercase p or uppercase P throughout and avoid mixing (l.35 lowercase, ll.38/40 uppercase).

2) Author Summary: l.56, please change to: “There is a lack…”

3) Author Summary: Under ‘What Do These Findings Mean?’, please swap the last two bullet points.

4) Figure 1: Please change “Shurugwi children who completed SHINE study” to “children who completed SHINE study in Shurugwi”.

5) Figure 1: Please change to “421 live births born to HIV-positive mothers in Shurugwi” and to “2174 live births born to HIV-negative mothers in Shurugwi”.

Before your manuscript can be formally accepted you will need to complete some formatting changes, which you will receive in a follow up email (including the editorial points above). Please be aware that it may take several days for you to receive this email; during this time no action is required by you. Once you have received these formatting requests, please note that your manuscript will not be scheduled for publication until you have made the required changes.

PRESS

Sincerely, 

Alexandra Tosun, PhD 

Associate Editor 

PLOS Medicine